# Cellular and molecular characterization of peripheral glia in the lung and other organs

**Shaina Hall[1], Shixuan Liu[2,3], Irene Liang[1], Shawn Schulz[1], Camille Ezran[2], Mingqian Tan** [1], **Christin S. Kuo** [1] *

1 Department of Pediatrics, Stanford University School of Medicine, Stanford, California, United States of America, 2 Department of Biochemistry, Stanford University School of Medicine, Stanford, California, United States of America, 3 Department of Chemical and Systems Biology, Stanford University School of Medicine, Stanford, California, United States of America

* ckuo@stanford.edu

**Data Availability Statement:** Data availability. The sequencing data that support the findings of this study are available in GEO under accession code GSE191178 and code generated to process data at https://github.com/liangirene/Tabula_Analysis.

## Abstract

Peripheral glia are important regulators of diverse physiologic functions yet their molecular distinctions and locations in almost all visceral organs are not well-understood. We performed a systematic analysis of peripheral glia, focusing on the lung and leveraging single cell RNA sequencing (scRNA-seq) analysis to characterize their cellular and molecular features. Using in vivo lineage studies, we characterized the anatomic, cellular, and molecular features of the *Sox10+* glial lineage of the mouse lung. Using high-resolution imaging, we quantified the distribution and cellular morphologies of myelinating, non-myelinating, satellite, and terminal glial cells with their intricate extensions along peripheral nerves, including terminals at specialized neurosensory structures within the lung. Spatial analysis of selectively expressed myelinating (periaxin/*Prx*, claudin 19/*Cldn*) or non-myelinating (sodium channel/*Scn7a*) glial cell genes identified by scRNA-seq analysis revealed molecularly distinct populations surrounding myelinated nerve fibers in the lung. To extend this analysis to primates and other organs, we extracted rare peripheral glial cells in whole organism scRNA-seq atlases of mouse lemur and human. Our cross-species data analysis and integration of scRNA-seq data of ~700 peripheral glial cells from mouse, mouse lemur, and human glial cells identified conserved gene expression of molecularly distinct peripheral glial cell populations. This foundational knowledge facilitates subsequent functional studies targeting molecularly distinct subsets of peripheral glia and integrating them into organ-specific disorders of autonomic dysregulation. In addition, our cross-species analysis identifying conserved gene expression patterns and glial networks in extrapulmonary organs provides a valuable resource for studying the functional role of peripheral glia in multiorgan human diseases.

## Introduction

Significant advances have been made in understanding the diverse roles of central nervous system glia, including regulating synaptic communication, neuronal excitability, neuronal

These gene count and metadata for existing data sets were derived from the following resources available in the public domain: Tabula Muris Senis [https://figshare.com/articles/dataset/Processed_files_to_use_with_scanpy_/8273102/2]; Tabula Microcebus [https://figshare.com/projects/Tabula_Microcebus/112227]; and Tabula Sapiens [https://figshare.com/projects/Tabula_Sapiens/100973].

**Funding:** This work was supported by funding from Stanford University COVID relief program, Department of Pediatrics Maternal and Child Health Research Institute (MCHRI) at Stanford University School of Medicine, and Chan Zuckerberg Foundation, Human Lung Cell Atlas (CZF2019-002438) funding to Mark Krasnow. C.S.K. is a Tashia and John Morgridge Endowed Faculty Scholar of the MCHRI. None of the material has been published or is under consideration for publication elsewhere. All animal husbandry, maintenance, and experiments were performed in accordance with Stanford University's IACUC-approved protocols (APLAC 32092).

**Competing interests:** The authors have declared that no competing interests exist.

differentiation (reviewed by Allen and Lyons) [1] and recently, in COVID neuropathology [2, 3]; however, relatively little is known about the distribution, and molecular subtypes of glia in almost all peripheral organs. While diverse roles of central nervous system glia have been established, the role of peripheral glia (also known as Schwann cells) and how they may differ across organ systems or by region within the same organ remains unknown. Existing data suggest peripheral glia are important regulators of diverse physiologic functions. For example, classic myelinating glial cells associated with myelinated axons, carry out homeostatic supportive functions and regulate post-injury inflammatory responses by coordinating macrophage recruitment [4, 5]. Beyond the supportive and reparative functions attributed to peripheral glia associated with nerve bundles [6, 7], the molecular profiles and functions of non-myelinating glia of major visceral organs are largely unexplored outside the enteric nervous system, where initial scRNA-seq profiling identified three transcriptionally-defined groups of glial cells [8] that require additional characterization to determine their distribution and function.

Here, we used two complementary approaches to characterize the molecular profiles of peripheral glial cells, focusing on the lung. Neural regulation of the pulmonary system includes autonomic inputs to control airway smooth muscle tone, submucosal gland secretion, and integrate information from sensory neurons innervating airways, vasculature, and neuroepithelial bodies [9–11]. A systematic analysis and understanding of the spatial and molecular heterogeneity of peripheral glial cells intimately associated with neural elements throughout the lung is an important initial step towards probing their specific respiratory functions. We first analyzed scRNA-seq of lung glia isolated by enriching for *Ascl1*-lineage-labeled cells to identify their molecular profiles. Next, to compare gene expression profiles of lung glia to peripheral glia of other organs, we analyzed existing large-scale whole organism single cell transcriptomic atlases for mouse [12], mouse lemur [13] and humans [14]. We identified glial cells from each species and organ through a systematic analysis to extract cells whose profiles were enriched in expression of both classical and new glial genes. Quantification of $Sox10^+$ glial cells revealed populations not previously visualized within the lung. This cellular and molecular analysis provides a foundation for studying their diverse predicted functional roles.

## Results

### Molecular profiles of lung glia identified by scRNA-seq

While non-myelinating glia have been identified in the lung by immunohistochemical detection of the classic glial markers, glial fibrillary acidic protein (GFAP) and S100B [15], they are not readily isolated and identified in mature lungs using routine single cell isolation strategies. Their small numbers in comparison to other cell types and close association with intricate neurons innervating the lung presented a challenge to isolate for single cell RNA sequencing studies [12, 16, 17]. We obtained glial cells for analysis by scRNA-seq using a similar genetic strategy to the one we previously described to label and enrich for pulmonary neuroendocrine cells (S1A Fig) [18]. Prior to embryonic day 13.5 (E13.5), we observed expression of the proneural transcription factor, Achaete-scute homolog 1 (*Ascl1*), in neural and glial progenitors within the lung. By inducing recombination at sequential embryonic developmental stages using $Ascl1^{CreER/+}$; $Rosa26^{LSL-ZsGreen1}$ mice [18], we showed neural and glial progenitors transiently expressed *Ascl1* in developing lung (Fig 1A–1C and S1B and S1C Fig). Among the *Ascl1*-lineage positive cells, glial cells formed a distinct cluster expressing the classic markers: *Gfap*, *S100b* (S100 calcium-binding protein), and *Plp1* (proteolipid protein 1) (Fig 1D and 1E). From this initial population of lung glial cells, we identified the genes *Sox10* (sex determining region Y box 10), *Gfra3* (GDNF family receptor alpha-3), *Kcna1* (potassium voltage-gated channel, Kv1.1), and *Cdh19* (Cadherin19), with glial cell selective expression and with greater

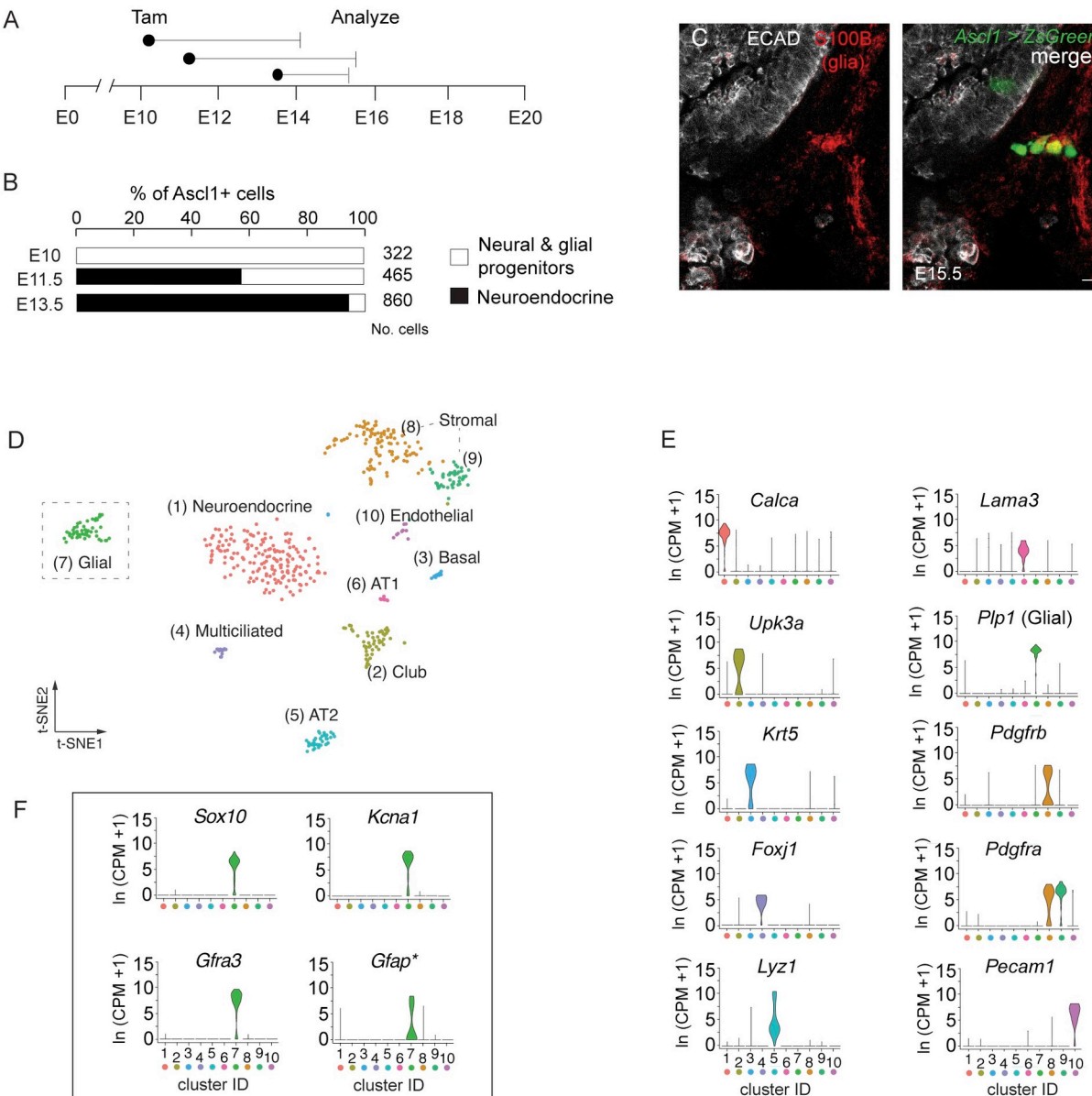

**Fig 1. Strategy for isolating glial cells and identification by scRNA-seq.** (A) Scheme for embryonic labeling of glial progenitors using *Ascl1*^CreER^; *Rosa26*^LSL-Zsgreen^ mice (*Ascl1 > ZsGreen1*) and inducing recombination by tamoxifen (Tam) injection at the indicated embryonic (E) days (black dots). Lungs harvested between E14 –E15.5 for analysis by immunohistochemistry (B) Quantification of lineage-labeled *Ascl1*^+^ progenitors when tamoxifen was administered at the indicated embryonic stages (E10, E11.5, and E13.5). (C) Confocal image of a single optical plane of a portion of embryonic mouse lung showing *Ascl1*-expressing progenitors within the epithelium and within neural progenitors. S100B expression in glial progenitors detected by fluorescence immunohistochemistry (red) Bar, 10 μm. (D) tSNE (t-distributed stochastic neighbor embedding) plot representing 534 mouse lung cells, including the population of enriched pulmonary neuroendocrine cells and other control cells we described previously (tSNE plot modified from Kuo et al, 2022 eLife). Glial cells were identified by their specific expression of classic glial genes (*Gfap, S100b, and Plp1*), but an analysis for marker genes identified additional genes (S1 Table) that were more broadly expressed in lung glial cells. (E) Violin plots showing expression of representative canonical markers for the other control cell types. Cluster identification (ID) corresponds to cell types indicated in panel D. Colored dots along x-axis correspond to the cell types in the following order from left to right: (1) Neuroendocrine, salmon color; (2) Club, olive; (3) Basal, blue; (4) Multiciliated, lavender; (5) Alveolar type 2 (AT2), teal; (6) Alveolar type 1 (AT1), pink; (7) Glial, green (8) and (9) stromal populations, brown and bright green, respectively; (10) Endothelial, lilac. (F) Violin plots showing expression of three of the most selective glial genes (*Sox10, Gfra3, and Kcna1*) identified by scRNA-seq in this data set compared to the classic marker *Gfap*, indicated by asterisk. Glial cell cluster (green arrowhead). Cluster IDs represented as above. Scale for all violin plots represent log-transformed unique molecular identifiers/(10,000 +1), [ln(UMI/10K+1)].

sensitivity compared to the classic marker, *Gfap* (Fig 1F, S1D Fig and S1 Table). The transcription factor, *Sox10*, is critical for the development of glia from the neural crest [7] and is also expressed in other peripheral glia [19, 20].

## Identifying molecular profiles of other peripheral glia by scRNA-seq analysis of a whole organism transcriptomic atlas of the mouse

To compare gene expression profiles of lung glia to peripheral glia in other organs, we analyzed an existing mouse single cell whole organism atlas, *Tabula Muris Senis*, comprising 356,213 cells captured from 23 different tissues. This revealed a cluster enriched for expression of a panel of classic pan-glial genes, including *Plp* and *Sox10*, which were among the most highly expressed among the classic genes. (Fig 2A, S2A and S2B Fig). We thus isolated the cells within this cluster and removed all central nervous system associated glia. Additional details are described in methods. Newly identified peripheral glial genes also showed selective expression within this cluster compared to cells across the entire atlas (S2C and S2D Fig), described further below. Within each tissue, we identified individual glial cells enriched for expression of classic glial marker genes (*Sox10*, *Ncam1*, *Plp1*, *S100b*, and *Gfap*) and filtered out any contaminating cells expressing classic epithelial, stromal, and immune marker genes (S2E–S2G Fig). We thus identified 435 peripheral glial cells (0.12% of cells) across seven tissues (limb muscle, heart, trachea, bladder, fat, lung, and kidney), though most originated from the limb muscle (n = 258) (Fig 2B, S2 Table). Given their relative sparsity, many of these glial cells were previously missed and annotated as non-glial cells. Clustering of cells from these seven tissues into the same UMAP embedded space revealed three transcriptionally distinct populations (Fig 2C and 2D) representing myelinating glial (MG), non-myelinating glial (NMG), as well as terminal glial (TG) cells which are a sub-population of non-myelinating glia, but unlike NMG cells, localized specifically to the neuromuscular junction [21, 22].

In addition to identifying pan-glial genes expressed by all glial cells, we also identified genes selectively expressed within each of the three populations (S3 Table). For example, *Cryab* was classified as a pan-glial gene because it was expressed in all three glial subtypes whereas periaxin (*Prx*) and claudin 19 (*Cldn19*) were selectively expressed by MG cells (Fig 2E and 2F, S2C and S2D Fig). Interestingly, *Cldn19*, was one of the most specific genes expressed in this cluster and was previously reported to localize to tight junction-like structures of peripheral MG cells. *Cldn19*-deficient mice also exhibited symptoms of peripheral neuropathy [23]. Among the NMG cells, *Ncam1* and *Scn7a* were expressed in a subset of NMG cells (Fig 2G). A distinct sub-population of NMG cells composed mostly of cells from limb muscle were labeled TG cells because they expressed butyrylcholinesterase (*Bche*), which has been previously reported to be expressed in TG cells at the neuromuscular junction (Fig 2H) [21, 22]. In some cases, genes were more selectively expressed in each class than previously reported classical genes. For example, *Cldn19*, *Prx*, and plasmolipin (*Pllp*), were more selective for myelinating glia than the classic myelinating glial cell genes myelin protein zero (*Mpz*) and myelin basic protein (*Mbp*) by scRNA-seq (Fig 2I). Since the glial cells we analyzed by scRNA-seq in mouse lung were enriched for expression of non-myelinating glial cell genes, we concluded those cells represented non-myelinating glia and additional spatial mapping below demonstrates their locations and molecular characteristics.

## Anatomic, cellular, and molecular features of lung glial cells

The anatomic, cellular, and molecular features of non-myelinating glia are largely unknown for most peripheral organs. To better characterize these cells within the lung, we used the expression profiles from both the Tabula Muris Senis and *Ascl1*-enriched lung scRNA-seq datasets

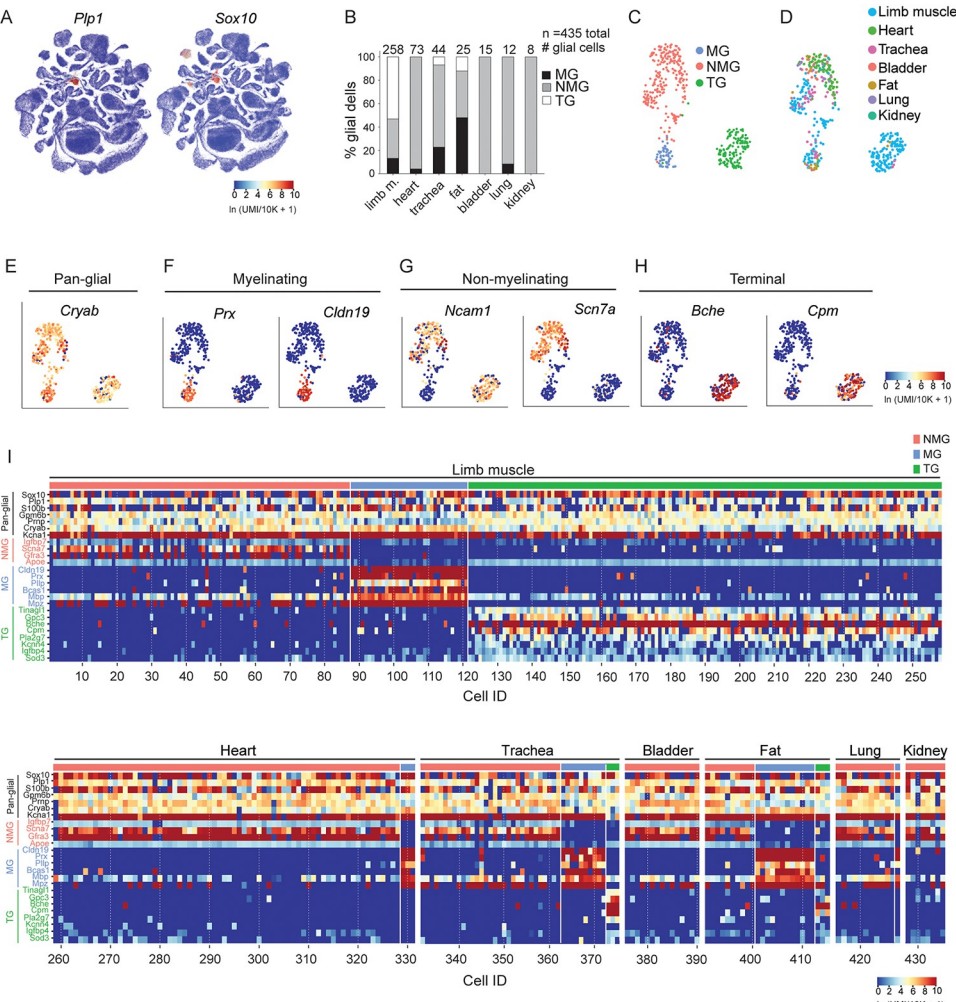

**Fig 2. Identification and analysis of peripheral glia reveals molecular profiles of myelinating, non-myelinating, and terminal glia in mouse.** (A) UMAP representation of all cells in the Tabula Muris Senis dataset showing proteolipid 1 (*Plp1* and *Sox10* highly expressed in the glial cell-enriched cluster. (B) Quantification of glial cells across Tabula Muris Senis. Number of glial cells isolated in each tissue indicated in parentheses: limb muscle, limb m. (258), heart (73), trachea (44), fat (25), bladder (15), lung (12), kidney (8). Glial cell type is indicated by shading: myelinating glial cell (MG, black), non-myelinating glial cell (NMG, grey), and terminal glial cell (TG, white) (n = 435 total glial cells). (C) Transcriptionally distinct Leiden clusters within peripheral glial cell population we identified in Tabula Muris Senis. MG (blue dots, n = 60 cells); NMG (pink dots, n = 232 cells); and TG (green dots, n = 143 cells) (D) Tissue or organ from which each cell was isolated. Note: terminal cells were largely isolated from limb muscle. (E) Feature plot showing expression of pan-glial marker, *Cryab* (F) Feature plots showing top differentially expressed genes in myelinating glial cells (e.g. *Prx* and *Cldn19*) (G) *Ncam1* and *Scn7a* expression in non-myelinating glial cell cluster (H) Feature plots showing terminal glial cells of neuromuscular junction express *Bche* and *Cpm* (I) Heatmaps showing gene expression levels of pan-glial (black lettering), non-myelinating (NMG, pink lettering), myelinating glial (MG, light blue lettering), and terminal glial (TG, green lettering) cells by scRNA-seq in each of the 7 indicated organs/tissues in mouse. Note: several myelinating glia genes (*Cldn19*, *Prx*, *Pllp*, *and Bcas1*) were more selectively expressed than the classic myelinating glial genes, myelin basic protein/*Mbp* and myelin protein zero/*Mpz*. Scale for all feature plots and heatmap represent log-transformed unique molecular identifiers/10,000 +1 [ln(UMI/10K+1)].

described above to predict that the *Sox10+* cell lineage would comprehensively label mouse pulmonary glia. To determine the number and distribution of lung glial cells, we used in situ single cell high resolution microscopy and quantified glial cells in *Sox10*^CreER; *Rosa26*^LSLTdt mice by systematic analysis of proximal to distal lung regions across entire lung lobes.

This revealed that mesh-like glial networks surrounded proximal airways such as the primary bronchus and conducting airways (Fig 3A, S3A and S3B Fig) but were absent from distal alveolar regions (S3C and S3C' Fig). Within the vasculature, glia were also absent from the accompanying pulmonary arteries, but instead formed networks surrounding pulmonary veins in mice (Fig 3A, S3D Fig). By immunostaining whole mount sections of lung, we found all (83/83, 100%) pulmonary vein glia associate with tyrosine hydroxylase (TH$^+$) nerve fibers, whereas only 67% (1087/1623) of airway glia associated with TH+ sympathetic nerve fibers (S3E and S3F Fig). Thus, the non-myelinating glia of pulmonary veins associated exclusively with noradrenergic neurons, while airway non-myelinating glial cells associated with additional neuronal subtypes.

We categorized lung glial cells into four classes, drawing upon classic descriptions of glia in other organs [24]. The four classes are as follows: myelinating glial (MG) cells that produce myelin fibers ensheathing axons and the other 3 sub-classes of non-myelinating glial cells, which do not produce a myelin sheath. We distinguish the non-myelinating glial (NMG) cells that surround airways and the pulmonary vein from the satellite glial cells (SGs), that surround neuronal soma within intrinsic ganglia of the lung and terminal glial (TG$^{NS}$) cells associated with neurosensory clusters (described below) (Fig 3A–3F). Non-myelinating glial cells (NMGs) that form airway and vascular networks were the most abundant population (n = 6949/9109 glial cells, 76%), while SGs (1.2%) and TG$^{NS}$ (0.31%) were much rarer (Fig 3B).

To visualize and analyze individual adjacent glial cells, we used either $Sox10^{CreER}$ to drive expression of a multicolor (rainbow/$Rbw$) Cre-responsive reporter $Rosa26^{LSL-Rbw}$ or $Gfra3^{CreER}$ in combination with the $Rosa26^{LSL-Tdt}$ reporter line to sparsely label glial cells in vivo. Myelinating glia (MG) surrounded axons, forming the classical support cells of large-diameter fast-conducting axons. Individual cells with elongated, fine extensions wrapped a single axon within nerve bundles. Myelin basic protein (MBP)-producing glial cells were oriented along axons traveling in nerve bundles parallel to conducting airways (Fig 3C). Individual non-myelinating airway and vascular glial cells often had multipolar projections compared to myelinating glial cells (Fig 3C and 3D). Single non-myelinating glial (NMG) cells had either unipolar, bipolar, or multipolar projections and most NMGs lining the airway and pulmonary vein were either bipolar or multipolar. For example, a single NMG cell with four projections extending from the soma ranging from 40 to over 100 μm in length is representative of the complex morphologies commonly observed (Fig 3G and S4A–S4E Fig). Multicolor labeling of individual glial cells within nerve bundles revealed characteristic patterns of cytoplasmic extensions. For example, the thin, ensheathing. extensions surrounding nerve fibers fit classic descriptions of myelinating glia, and two examples of lung myelinating glial cells (MG1/mCherry and MG2/ECFP, Fig 3H–3J) exhibit this feature. Other glial cells had long, cellular extensions without forming ensheathing structures and exhibited the morphologies of airway non-myelinating glia. A representative non-myelinating glial cell (NMG1/mOrange) is shown (Fig 3K and 3L).

Existing data on SG cells of autonomic ganglia are primarily derived from studies of sympathetic ganglia (reviewed by Hanani and Spray) [25], but little is known about the intrinsic ganglia of the lung and their associated SG cells. Here, we observed flattened glial cells surrounding individual neuronal soma of intrinsic ganglia (Fig 3M and 3N) in the lung. Individual SG cells formed a dome-like structure covering ~50% of the surface area of the neuronal soma within parasympathetic ganglia, and some extended projections along the associated axons (Fig 3N, S1 Movie). Finally, we identified NMG cells associated with sensory nerve terminals [9–11] and clusters of neurosensory cells called neuroepithelial bodies (NEBs). We called these terminal neurosensory (TG$^{NS}$) glia to distinguish them from terminal (peri-synaptic) glial cells at neuromuscular junctions [26–28] (Fig 3F). TG$^{NS}$ glial cells either contacted NEBs at the basal surface (Fig 3O) or extended projections that penetrated the basement

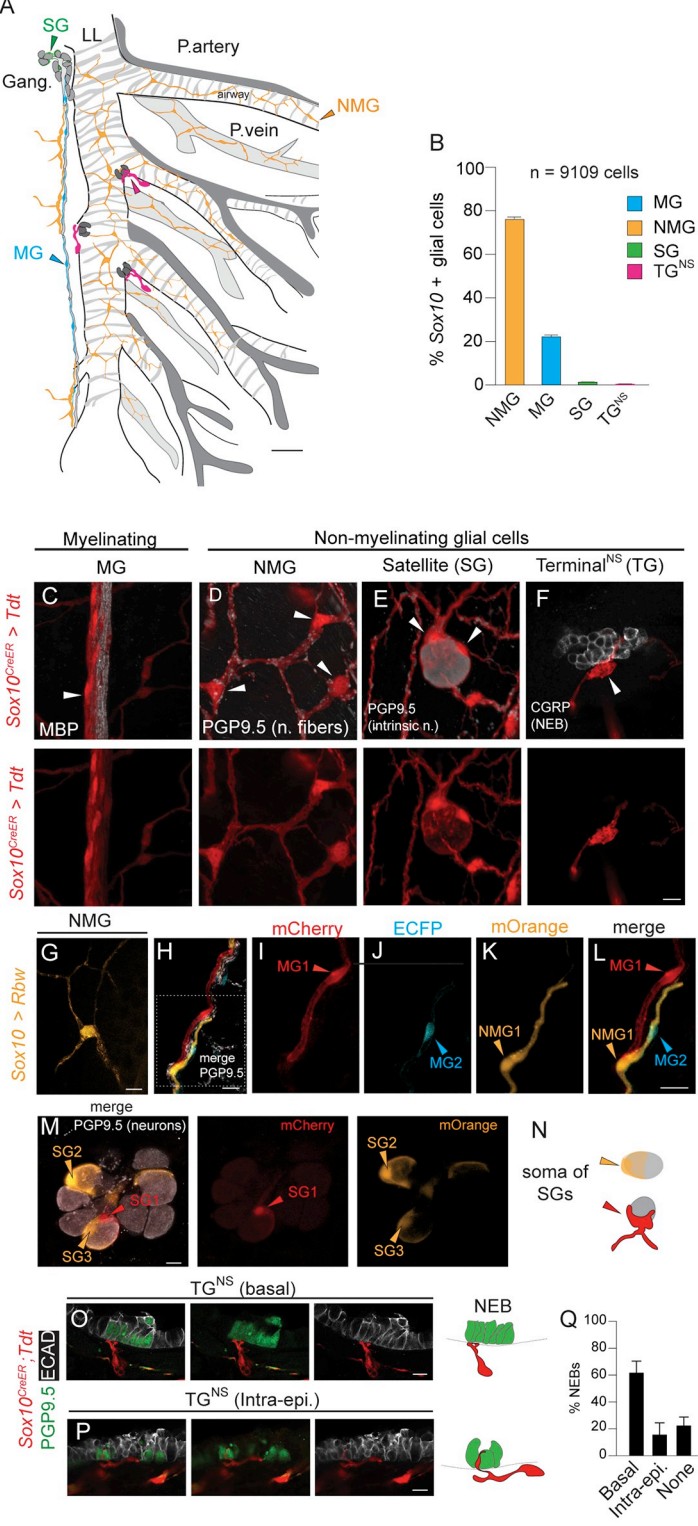

**Fig 3. Anatomic and cellular features of lung glial cells.** (A) Schematic of left lobe (mouse) showing the primary left main bronchus (LL) and lateral branches 1–4 (L1-4) as indicated. The pulmonary artery (P. artery, dark grey shade) for L1 is labeled and the subsequent lateral branches with accompanying pulmonary arteries are shown. The pulmonary vein (P.vein, light grey shade) is located between the airway branches. The airways and pulmonary veins are surrounded by a network of non-myelinating glial (NMG cells, light orange). Note the pulmonary arteries are not

surrounded by glia. The myelinating glial cells (MG, blue shade) ensheath myelinated nerve fibers most readily observed along the airways. Satellite glial cells (SGs, green shade) surround intrinsic neuron ganglia (Gang.) that were largest in the proximal airway regions. Finally, terminal cells adjacent to or contacting neuroepithelial bodies (NEBs) are found at branch points and called terminal neurosensory glial (TG$^{NS}$) to distinguish them from terminal glia at neuromuscular junctions. An examples of each glial cell class is indicated by a filled arrowhead with the corresponding color (MG/blue; NMG/orange; TG/green; TG$^{NS}$/magenta). This schematic represents the anatomic locations of each class but not the density of individual glial cells. Scale. 200 μm. (B) Quantification of *Sox10+* glial cells by class. Total number of cells by class as follows: NMG (6949/9109, 76%), MG (2025/9109,22%), SG (107/9109, 1.17%), TG$^{NS}$ (28/9109, 0.31%). PN 71, 74, 91 days, n = 3 mice. Error bars, 95% C.I. indicated. (C) Confocal images of glia visualized by in vivo labeling in *Sox10*$^{CreER}$; *Rosa26*$^{LSL-Tdt}$ mice (PN221 days old) are presented in panels C-F. Panel C shows myelinating glia form parallel bundles along nerve fibers immunostained for myelin basic protein (MBP). (D) Close-up region of the NMG network along airways with cell bodies of 3 NMG cells (white arrowheads) associated with neuronal fibers (protein gene product 9.5; PGP9.5, white). (E) A single lung intrinsic neuron with the soma of 2 SG cells (white arrowheads). (F) A neuroepithelial body (NEB) visualized by fluorescent immunohistochemistry for the peptide, calcitonin-gene-related peptide (CGRP). Associated terminal neurosensory glial cells (TG$^{NS}$) penetrate the basement membrane and have intra-epithelial contacts on epithelial cells. Scale bar, 10 μm. (G) Labeling of a single non-myelinating glial cells along the airway using *Sox10CreER* mice in combination with the *Rosa26LSL-Rbw* multicolored reporter (*Sox10 > Rbw*) shows four projections extending from the soma ranging from 40 to over 100 μm in length (Scale bar, 10 μm). (H) Maximum intensity projection demonstrating myelinating glial cells along the airway of *Sox10*$^{CreER}$; *Rosa26*$^{LSL-Rbw}$ mice (*Sox10 > Rbw*). (I) The soma of a single myelinating glial cell (MG1, red arrowhead) has bipolar extensions (expressing the membrane cherry/mCherry, fluorescent protein). (J) The soma of a second MG cell (MG2 cell soma, cyan arrowhead) expresses enhanced cyan fluorescent protein/ECFP and has similar projections that form a sheath surrounding an associated nerve fiber (protein gene-product 9.5/PGP9.5, white). The total length of MG1 is ~150 μm. The total length of MG2 is ~100 μm. (K) A single adjacent non-myelinating glial cell (NMG1) expressing membrane orange (mOrange) has extensions that do not ensheath the nerve fiber. (L) Merged panel showing multicolored labeling of 2 MG cells and a single NMG cell adjacent to a nerve fiber (M) Close-up maximum intensity projection confocal image of an intrinsic ganglion and 3 surrounding satellite glial cells (SGCs). Respective soma of intrinsic neurons detected by expression of the pan-neuronal marker, protein gene product 9.5 (PGP9.5, white). Scale 10 μm. Adjacent panels show individual satellite cells visualized in *Sox10 > Rbw* mice. A single satellite glial cell forms a goblet-like structure surrounding the neuronal soma and extends a projection along the proximal intrinsic neuron nerve fiber (SG1, red arrowhead). Two satellite glial cells (SG2, SG3) expressing the fluorescent protein, mOrange, form dome-like flattened cells covering ~30–50% of the surface area of each neuron cell body (PGP9.5, grey ovals). Orange and red arrowheads point to SG soma. Scale 10 μm. (N) Schematic to right illustrates morphology of individual satellite glial cell (SGLs). (O) Terminal cells (TG$^{NS}$), *Sox10 CreER > Tdt* (red) associated with nerve terminals at the basal side of a neuroepithelial body (NEB). NEBs visualized by fluorescence immunohistochemistry (PGP9.5, green), Scale 10 μm. Schematic to right (dotted line, basement membrane) (P) Glial cells with extensions that penetrate beyond the basement membrane to associate sensory nerve terminals contacting neuroepithelial bodies (NEBs). E-cadherin (ECAD, white). Scale 10 μm. (Below) Schematic to right (dotted line, basement membrane) (Q) Quantification of glial cell contact patterns at NEBs in mouse lungs: # NEBs with terminal glial cell contact at the basal surface (Basal, n = 94/151, 62%); intra-epithelial (Intra-epi., n = 24/151, 16%); or with no glial cell contact (None, n = 33/151, 22%); n = 4 mice. Statistically significant differences were observed between Basal vs. Intra-epi. (p<0.0001) and Basal vs. none (p<0.0001). 2-proportion z test. Error bars, 95% C.I. indicated.

membrane along with a NEB-innervating nerve terminal (Fig 3P, S2 Movie). Most NEBs (118/151, 78%) were associated with glial cells either at the basal surface (94 of 151, 62%) or via intra-epithelial cellular projections (24/151, 18%). Only 22% (33/151) had no glial association (Fig 3Q). We conclude lung glia exhibit diverse cellular morphologies and their close association with sensory structures in addition to NMG cell networks surrounding airways and vasculature suggests multiple physiologic functions.

## In situ spatial and molecular analysis of lung glial cells

In addition to identifying non-myelinating glial cell networks along the airways and vasculature previously identified by classical glial markers GFAP and S100B [15], we discovered glial cells that lacked GFAP expression under normal conditions. Using *Sox10*$^{CreER}$; *Rosa26*$^{LSL-Tdt}$ mice to visualize lung glia and co-localizing GFAP expression by immunohistochemistry, we found only 40% (142/369) of non-myelinating glia were GFAP+ (Fig 4A and 4B), consistent with our scRNA-seq data (S1D Fig). The GFAP- glial cells along the airway were integrated into the network of NMG cells. One of the most broadly expressed NMG cell-selective genes

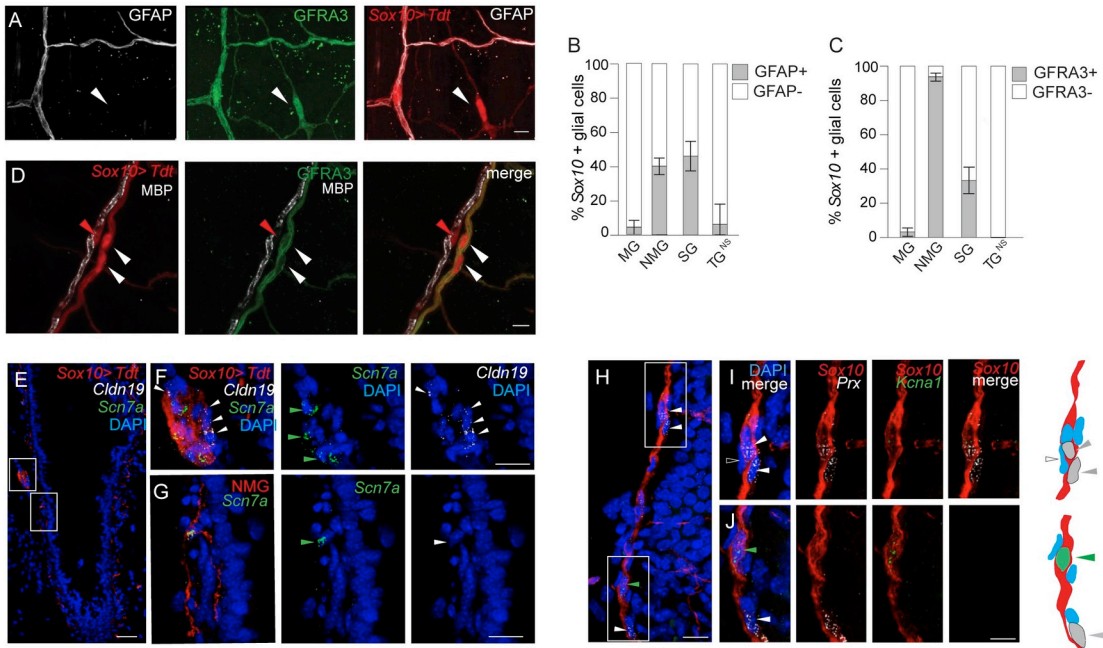

**Fig 4. In situ identification of molecularly distinct myelinating and non-myelinating lung glia.** (A) Close-up confocal images of glial cells visualized by fluorescence immunohistochemistry to detect the classic marker, glial fibrillary acidic protein (GFAP, white) co-expressed in $Sox10^{CreER} > Tdt$ labeled glial cells (red) along the airway; GDNF family receptor alpha 3 (GFRA3, green), white arrowhead points to an example of GFRA3+, $Sox10+$ (but GFAP-negative) non-myelinating glial cell along airway. Bar, 10 μm. (B) Quantification of $Sox10+$ glial cells co-expressing GFAP by fluorescence immunohistochemistry for each glial class. Total number of non-myelinating (NMG) cells (n = 359), myelinating (MG) (n = 96), satellite (SG) (n = 128), and terminal (TG[NS]) (n = 16). GFAP expression by class as follows: NMG (143/359, 40%); MG (4/96, 4%); SG (59/128, 46%); terminal[NS] (TG[NS]) cells (1/16, 6%). PN 80,100 days, n = 2 mice. Statistically significant differences were observed between NMG vs. MG (p<0.0001), NMG vs. TG[NS] (p<0.01), MG vs. SG (p<0.0001), and SG vs. TG[NS] (p<0.005). 2-proportion z test. Error bars, 95% C.I. indicated. (C) Quantification of $Sox10+$ glial cells co-expressing GFRA3 by fluorescence immunohistochemistry for each glial class. Total number of NMG (n = 475), MG (n = 218), SG (n = 144), TG[NS] (n = 33). GFRA3 expression by class as follows: NMG (447/475, 94%); MG (7/218, 3.2%); SG (48/144, 33.3%). PN 25, 80, 100 days, n = 3 mice. Statistically significant differences were observed between NMG vs. MG (p<0.0001), NMG vs. SG (p<0.0001), NMG vs. TG[NS] (p<0.0001), MG vs. SG (p<0.0001), and satellite vs. terminal (p<0.001). 2-proportion z test. Error bars, 95% C.I. indicated. (D) Confocal image of MG cell with extensions ensheathing associated nerve fiber. MG cell adjacent to a myelinated nerve fiber expressing myelin basic protein (MBP). Soma of MG cell indicated by red arrowhead. Two neighboring non-myelinating glial cells ($Sox10$-expressing) are GFRA3+ by fluorescence immunohistochemistry (green) and MBP-negative. Soma of NMG cells indicated by white arrowheads. Scale, 100 μm. (E) Confocal images showing multiplex single molecule in situ fluorescence hybridization of a section of an airway expressing the MG cell gene, claudin 19/ $Cldn19$, within ($Sox10 > Tdt$, red) lineage labeled glial cells surrounding large caliber nerve fibers (white arrowhead). Lineage-labeled $Sox10+$ glial cells are detected by immunostaining for the red fluorescent protein in panels E-J below. The sodium channel ($Scn7a$) was expressed in NMG cells (green arrowhead) in addition to a subset of glial cells adjacent to myelinated fibers. DAPI nuclear stain, blue. Scale bar, 50 μm. (F) Close-up image of the upper boxed region in panel E. (G) Close-up image of lower-boxed region in panel E. Note non-myelinating glial cell (green arrowhead) has multipolar projections characteristic of airway NMG cells (Fig 4E). DAPI nuclear stain, blue. Scale, 20 μm. (H) Multiplex single molecule in situ fluorescence hybridization of a separate intrapulmonary airway with adjacent nerve fiber and lineage-labeled glial cells from $Sox10 > Tdt$ mice ($Sox10$, red) as in panels above. Periaxin ($Prx$, white); $Kcna1$, green. Scale bar, 20 μm. (I) Close-up image of upper boxed region in (H). Two $Prx+$ myelinating glial cells situated along the nerve fiber are indicated (white arrowhead). Note there is a third cell in this field that is not expressing $Prx$. Scale, 10 μm. (J) Close-up image of lower boxed region in (H). Example of a glial cell expressing the potassium channel, $Kcna1$, but not $Prx$ indicated by green arrowhead) and just below it, another myelinating glial cell expressing $Prx$, but not $Kcna1$ (white arrowhead). Scale, 10 μm. Accompanying schematics, right panels. Glial fibers (red) Nuclei (blue oval), MG cell soma (grey oval) expressing $Prx$ (grey arrowheads), glial cell soma (green oval) expressing $Kcna1$ (green arrowhead). Note an adjacent nuclei express neither $Prx$ nor $Kcna1$ (open arrowhead).

identified by scRNA-seq analysis of lung glia, was the neurotrophic factor family receptor 3 (*Gfra3*) (S1D Fig). Thus, we investigated GFRA3 expression by immunohistochemistry and found almost all NMG cells (447 of 475 cells, 94%) were GFRA3+. In contrast, MG cells showed little to no GFRA3 expression (7 of 218 cells, 3.2%) (Fig 4A and 4C). Thus, GFRA3 expression distinguishes non-myelinating from myelinating glia in mouse lung both by scRNA-seq and by immunohistochemistry, except for a subclass of NMG cells found at neuro-sensory nerve terminals (TG^NS) with cuboidal cell soma and were GFRA3-negative (S4F Fig). In contrast, the slender, multipolar NMG cells surrounding airways and pulmonary veins had high levels of GFRA3 expression (Fig 4A). These results demonstrate NMG subclasses have distinct cellular and molecular features.

To determine whether glial cells expressing MG and NMG genes identified by scRNA-seq analysis localized to distinct glial cells in the lung, we performed multiplex single molecule in situ hybridization studies. Claudin 19 (*Cldn19*) and the sodium channel gene (*Scn7a*) were selectively expressed in myelinating and non-myelinating glia, respectively by scRNA-seq. In situ analysis demonstrated *Cldn19* expression in glial cells surrounding myelinated large caliber nerve fibers adjacent to the conducting airways (Fig 4E–4G) and no expression was detected in NMG glial cells that form a network along the airways. A single lineage-labeled non-myelinating glial cell expressing the fluorescent protein, Tdt, highlights the complex morphology of non-myelinating glia with multipolar extensions. *Scn7a* expression was detected in the cell soma. Interestingly, we also observed *Scn7a*-expressing cells adjacent to the soma of myelinating (*Cldn19*+) cells surrounding the large, myelinated fiber tract (Fig 4F). These results confirm the molecularly distinct glial populations observed by scRNA-seq analysis and we could demonstrate their expression in distinct glial cell populations in the lung corresponding to myelinating and non-myelinating populations. Interestingly, *Scna7a* was expressed in a subset of *Cldn19*-negative glia surrounding large fiber myelinated tracts (Fig 4G), suggesting molecular diversity among the glial cells surrounding myelinated nerve fibers with many glial cells sharing transcriptional profiles of non-myelinating glial cells. Another MG gene, periaxin (*Prx*) identified by scRNA-seq was indeed expressed in individual glial cells along intrapulmonary nerve fibers in locations consistent with their functional role in myelination (Fig 4H–4J). However, not all cells along the fiber expressed *Prx*. In the same sections, we also analyzed expression of the gene encoding the potassium channel (*Kcna1*), which was broadly expressed in the lung glial cells obtained by scRNA-seq. We indeed found expression in both non-myelinating glial cells and in some of the glial cells adjacent to myelinating glial cells as predicted (Fig 4J). This molecular characterization of lung glia using sensitive glial markers to identify diverse and complex cellular morphologies of glial cells associated with neurons and their terminals, including those associated with specialized neurosensory structures in the lung establishes a framework for investigating their roles in facilitating neuronal signaling between epithelial, vascular, and stromal compartments of the lung.

## Cross-species glial cell integration identified conserved peripheral glial cell gene expression

To extend our findings in mouse to primates, we next analyzed glial cells from Tabula Microcebus Murinus, a single cell whole organism transcriptomic cell atlas of the mouse lemur primate model [13]. We annotated glial cells from the mouse lemur as previously described [13] and identified gene expression profiles of myelinating (n = 47) and non-myelinating (n = 139) glial cells (S5 Fig and S4 Table) in mouse lemur across 8 different peripheral tissues: bladder, bone, fat, kidney, lung, pancreas, small intestine, and tongue (Fig 5A and 5B). Expression of both known glial genes (*SOX10*, *PLP1*, and *S100B*) as well as newly identified ones (*GPM6B*,

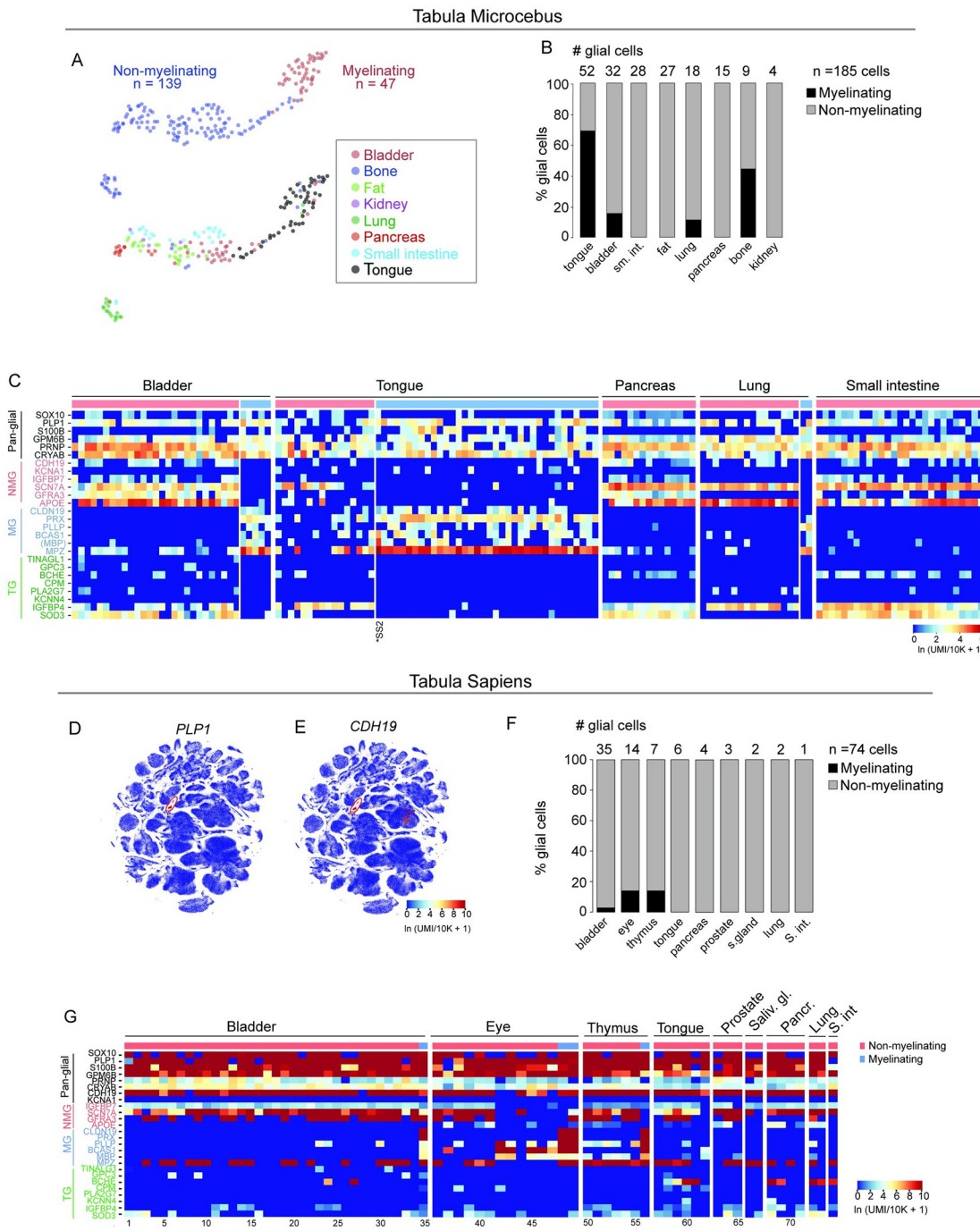

**Fig 5. Identification and analysis of peripheral glia in whole organism scRNA-seq atlases of primates.** (A) Umap representation of glial cell populations across the mouse lemur cell atlas [13]. Myelinating glial cells (n = 185), non-myelinating glial cells (n = 47 cells) (upper plot) and the 8 organs and tissues they were isolated from (lower plot). (B) Quantification of glial cells across Tabula Microcebus Murinus (referred to as Tabula Microcebus). Number of glial cells isolated in each tissue indicated in parentheses: tongue (52), bladder (32), small intestine, sm. int (28), fat (27), lung (18), pancreas (15), bone (9), and kidney (4). Total number of glial cells (n = 185 cells). Glial cells in Tabula Microcebus were identified and annotated by systematically curating genes both classic and new reported in peripheral glia. Interestingly, the NMG cells from tongue in the Tabula Microcebus dataset formed a continuum in the UMAP projection (panel A) and this result is consistent with co-expression of MG genes as shown in the next panel (C) Heatmaps showing gene expression levels of pan-glial (black lettering), non-myelinating (NMG, pink lettering), myelinating (MG, light blue lettering), and terminal (TG, green lettering) glial cell genes expressed by scRNA-seq in each of the 5

indicated organs/tissues the mouse lemur atlas (Tabula Microcebus). Cell isolated by SmartSeq2 protocol (*SS2). These organs were shown here because glial cells were also isolated from the same organs in the scRNAseq atlas of humans (Tabula Sapiens). The gene expression level for pan-glial markers is lower in the Tabula Microcebus data compared to Tabula Muris Senis and Tabula Sapiens data sets. These differences may reflect technical variations in sample preparation for scRNA-seq. Lower # read counts/glial cell in the lemur data (median: 2112; $10^{th}$– $90^{th}$ percentile range: 1097–5430 reads/cell) could result in higher gene "drop out" rate compared to mouse (median: 3715 reads/cell; $10^{th}$– $90^{th}$ percentile range: 2691–229491 reads/cell) and human data (median: 4155 reads/cell; $10^{th}$– $90^{th}$ percentile range: 2844–18263 reads/cell). The number of read counts per glial cell are listed in S2 and S5 Tables for mouse and human, respectively. Lemur read counts for each glial cell included in S6 Table are previously reported (Tabula Microcebus Consortium, 2024 https://www.biorxiv.org/content/10.1101/2021.12.12.469460v3). (D-E) UMAP representation across entire scRNA-seq human atlas showing expression of classic glial gene *(PLP1)* enriched in the indicated cluster (red oval) and newly identified gene, *CDH19* (E) highly enriched in human glial cells. (F) Quantification of glial cells from 7 different organs and the abundance of MG vs NMG in each organ are listed in panel. Note: we observed the largest numbers of cells from the large intestine (L. intestine) and small intestine (S. intestine). However, they were from a single donor (TSP14) (see methods section and github link). While these cells were excluded from subsequent analysis here, they likely represent subpopulations of peripheral glia and will be analyzed together with additional cells from other patients when data are available. (G) Heatmaps showing gene expression levels in each cell across the 5 indicated organs/tissues in human. Pan-glial genes (black lettering), non-myelinating genes (NMG, pink lettering), myelinating (MG, light blue lettering), and terminal glial genes (TG, green lettering). Expression of the genes, *MPZ* and *MBP*, were not selective for the MG cells in human. Scale for all feature plots and heatmap represent log-transformed unique molecular identifiers/10,000 +1 [ln(UMI/10K+1)].

*CDH19*, and *CRYAB*) were conserved between mouse and lemur glial cells. The most selectively expressed NMG cell genes, *SCN7A*, *APOE*, and *CHL1*, were also conserved (Figs 2, 5C and S2 Fig).

We then analyzed the Tabula Sapiens single cell transcriptomic atlas comprising nearly 500,000 cells from 24 human tissues and organs. Using a combination of pan-glial genes, we identified 74 peripheral glial cells across 8 human organs and excluded cells that co-expressed classic genes of other cell types (e.g. endothelial, stromal, epithelial) (Fig 5D–5G and S6A and S6B Fig). Again, the best pan-glial genes described above and MG and NMG glial genes across the three species were conserved (Fig 5G). For example, MG cell genes periaxin/*PRX*, plasmo-lipin/*PLLP*, brain-enriched myelin associated protein/*BCAS1*, *and CLDN19* expression were also detected in human. For comparison, glial cells isolated from 5 tissues and organs in both humans and mouse lemur are shown in Fig 5C and 5G. Although small numbers of MG cells were identified from human tissues, they could be distinguished by expression of a combination of the indicated MG genes.

In total, we identified 695 glial cells comprised of 435 mouse, 186 mouse lemur [13], and 74 human glial cells from the scRNA-seq atlases of mouse (Tabula Muris Senis) [12], mouse lemur (Tabula Microcebus) [13], and human (Tabula Sapiens) [14] representing 15 different organs and tissues. The tissue source and glial subtype for each cell is summarized, including our annotations of peripheral glia (S6 Table). To facilitate subsequent cross-species analysis and comparison, we integrated the scRNAseq data of 695 glial cells across all three. We first aligned genes according to orthology assignments from both the NCBI and Ensembl orthology databases, as described [13]. This resulted in a total of ~15,000 one-to-one orthologs across the combined data. Next, we applied Portal [29] to integrate and embed the data into a common UMAP (S6 Table) and found that individual cells, irrespective of species, formed three major clusters consistent with MG, NMG, and TG subtypes. Some glial cells (gray shade in panel A) were unclassified because they had hybrid gene expression patterns, and future studies are required to distinguish additional heterogeneity (Fig 6A–6C). SG cells are extremely rare and difficult to isolate without targeted enrichment so not represented here.

We identified MG, NMG, and TG cell selective gene expression within each cluster of the integrated glia data set which were conserved across all three species (Fig 6D–6G, S7–S10 Tables). For example, periaxin/*PRX*, claudin 19/ *CLDN19*, dystrophin-related protein 2/*DRP2*, and plasmolipin/*PLLP* were highly and selectively expressed in the MG cluster (Fig 6D). The

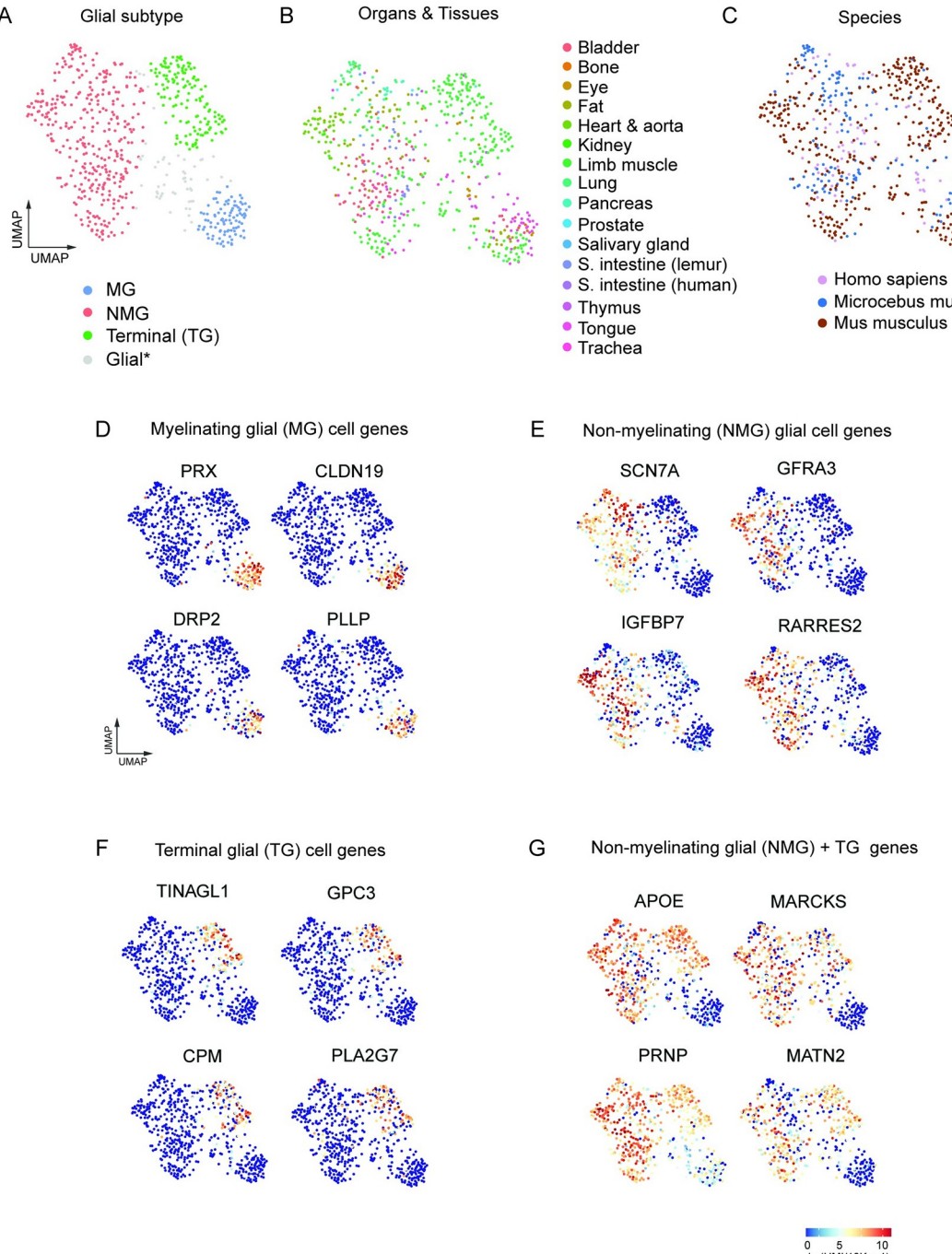

**Fig 6. Cross-species analysis of integrated scRNA-seq data from peripheral glia in whole organism atlases of mouse, mouse lemur, and human.** (A) UMAP plot of integrated scRNA-seq data from peripheral glia cells) n = 695 cells identified in Tabula Sapiens, Tabula Microcebus, and Tabula Sapiens (metadata summarized in S6 Table). Three transcriptionally distinct clusters representing MG, NMG, and TG cells. The *Glial cluster cells (grey dots) represent peripheral glial cells not further classified because they exhibited hybrid gene expression patterns that await future studies. (B) Glial cells from 15 organs and tissues across the 3 species. (C) UMAP plot showing integrated glial cells from *Mus musculus*, *Microcebus Murinus*, and *Homo sapiens* scRNA-seq atlases. (D)–(F) Feature plots showing conserved differentially expressed genes in the MG, NMG, and TG cells, respectively. (G) Feature plot showing differentially expressed NMG genes (including TG cells). Complete set of differentially expressed genes in the integrated glial dataset corresponding to panels D—G are listed in S7–S10 Tables. Values for heatmap scale, log-transformed unique molecular identifiers/10,000 +1 [ln(UMI/10K+1)].

NMG genes (*SCN7A*, *GFRA3*, insulin-like growth factor-binding protein 7/*IGFBP7*, and retinoic acid receptor responder protein 2/*RARRES2*) were expressed in NMGs, but not in TG cells (Fig 6E and 6F). The TG cells formed a separate cluster enriched for expression of *TINAGL1*, *GPC3*, *CPM*, *AND PLA2G7* (Fig 6F). NMG (including terminal glia) could be distinguished from MG cells by expression of apolipoprotein E/*APOE*, myristoylated alanine rich protein kinase C substrate/*MARCKS*, and the extracellular matrix protein gene matrilin 3/*MATN3* (Fig 6G). These glial cells represent a small fraction of the entire population within each organ and are challenging to capture; thus, there is likely greater diversity than represented here, especially among the NMG population within each organ. The complete list of genes distinguishing MG, NMG, and TG, and genes broadly expressed in all NMG (including TG) is reported in S7–S10 Tables.

The cross species integrated data of peripheral glia across organs identified valuable candidate genes for subsequent functional studies. For example, an immunomodulatory gene, apolipoprotein E (*APOE*), was one of most highly expressed genes distinguishing non-myelinating from myelinating glial cells across all three species (S7–S7C Fig). While prior studies support a significant role for *Apoe* in the pathogenesis of airway inflammatory responses in mice [30, 31] the molecular identity of cell type(s), including those expressing *Apoe* under normal conditions, were not known. Thus, we searched the mouse lung cell atlas (n = 6315 cells) for *Apoe* expression and identified the following additional cell types: interstitial macrophages, basal cells, and stromal cells (alveolar fibroblasts, adventitial fibroblasts, and myofibroblasts) (S7B Fig). Whole animal *Apoe*-deficient mice have pro-inflammatory signaling and pathologic remodeling of lung parenchyma and vasculature [32, 33]. Broad and conserved *APOE* gene expression in mouse, mouse lemur, and human NMG cells provides a new and conserved cellular target for subsequent studies elucidating their potential anti-inflammatory role following lung injury.

Conventional single cell dissociation and isolation strategies capture only small numbers of peripheral glial cells per organ. Thus, integrating and analyzing glial cell data across organs and species is a valuable strategy for identifying conserved genes that will facilitate developing new disease models in each organ that incorporates glial cells. The largest number of glial cells in the human atlas came from the bladder and we consistently identified glial cells in the three atlases analyzed (17, 32, and 35 bladder glial cells in Tabula Muris Senis, Tabula Microcebus, and Tabula Sapiens, respectively) as shown (Figs 2I, 5B and 5G). We hypothesized the bladder may be enriched for glial cells because the detrusor muscle of the bladder wall is extensively innervated and highly regulated to allow bladder filling when empty and timed micturition when full [34]. Indeed, there was a dense network of $Sox10^{CreER}$; $Rosa26^{LSL-Tdt}$ labeled NMG cells on the surface of mouse bladder wall (S8A Fig). The bladder wall (detrusor) muscle and the lamina propria (subepithelial stromal layer) glial cells had complex cellular morphologies resembling those of NMG cells we observed in lung and expressed *Sox10* and *Gfra3*, but not the classic glial gene, *Gfap*, by scRNA-seq and immunohistochemistry (S8B Fig). Within the detrusor muscle layer, only 37% (n = 69/186 cells) expressed GFRA3, suggesting additional diversity within the NMG cells here (S8E Fig). The $Sox10^{CreER}$; $Rosa26^{LSL-Tdt}$ lineage + cells of the lamina propria layer did not express myelin basic protein (MBP) and the few that did were found along large nerve fiber bundles innervating the bladder, consistent with the appearance of classic MG cells (S8C and S8D Fig). We conclude that glial cells also form a prominent network within the bladder with additional molecular diversity within the NMG glial cells. Integrating peripheral glia into models of neural and myogenic bladder control are important to address the high burden of physiologic bladder dysfunction due to trauma, inflammation, or degeneration [35, 36].

In this study, we leveraged existing data sets to identify conserved expression of MG, NMG, and terminal glial cells that will facilitate identification and incorporation of peripheral glia into disease models of diverse tissues. We focused on a systematic quantification and spatial analysis of peripheral glia in the lung and combined this understanding with the scRNA-seq analysis of lung glial cells. These foundational studies will facilitate studies targeting the physiologic contributions of molecularly distinct peripheral glial cells.

## Discussion

Peripheral glia are implicated in diverse physiologic processes and disease biology, and understanding their distribution and molecular subtypes within tissues of major visceral organs establishes a framework for integrating data derived from classic physiologic studies. Here, we present a systematic cellular and molecular characterization of lung glial cells and an analysis of integrated peripheral glial cell data from scRNA-seq whole organism transcriptomic atlases of mouse, mouse lemur, and human. Identifying conserved expression of functional glial genes will facilitate modeling human diseases in animals. In addition, this analysis of large-scale datasets provides a platform for probing the contributions of peripheral glia to a variety of conditions, including autonomic dysregulation within peripheral organs and provides novel approaches to understanding systemic inflammatory conditions where peripheral glia contribute to pathology. For example, in the classic demyelinating condition, Guillain-Barre syndrome, respiratory failure requiring mechanical ventilation occurs in ~30% of patients. While the major etiology of acute respiratory failure is attributed to progressive muscle weakness of respiratory muscles [37], the pathophysiology of persistent dyspnea and chronic respiratory insufficiency in some patients may be related to organ-specific glial dysfunction. Respiratory failure is one of the most serious complications, however, given the multisystemic pathology of this and other diffuse demyelinating conditions, our cross-organ and species approach establishes a valuable platform to study shared glial cell function across organs. For instance, expression of the voltage-gated sodium channel gene, *SCN7A*, is conserved across species and is highly expressed in non-myelinating glial cells across multiple organs. *SCN7A* encodes for an 'atypical' sodium channel that is activated in response to changes in extracellular sodium concentration. It is proposed to have a role in regulating local sodium homeostasis critical for neuronal health and signaling [38].

In addition to common glial characteristics, we also noted organ-specific differences in glial cell arrangement and distribution. For example, the population of morphologically distinct TG[NS] glial cells extending projections into neuroepithelial bodies (clusters of innervated sensory and secretory cells), and the satellite glial cells associated with lung intrinsic ganglia, are both extremely rare, comprising only 0.31% of total glial cells in the lung (Fig 3B). We speculate the TG[NS] population may form a neuro-glial sensory complex analogous to the glial cells, which have been identified at the subepidermal border within the skin forming a nociceptive 'glio-neural' complex [39]. These specialized cutaneous glial cells elaborate extensions forming a mesh-like network and transmit nociceptive signals to associated unmyelinated sensory nerve terminals. An important future question is to determine whether the morphologically distinct lung terminal glial cells may also be distinguished molecularly, and whether they contribute to perception of sensory information from the lung. Emerging data from scRNA-seq analysis of peripheral glia from other specialized sensory structures demonstrated additional molecular diversity and functional specialization. For example, satellite glia from sensory ganglia of the auditory system (spiral ganglia) express myelination genes, which are not expressed in satellite glia cells of the dorsal root ganglia [40].

Beyond the respiratory system, we also examined the distribution of *Sox10*-expressing cells in the bladder because of its extensive innervation. We observed the bladder is also surrounded by an extensive glial network and *Sox10* expression is associated with worse prognosis in bladder cancer [41], thus highlighting peripheral glial cells as important candidates for future investigation and integration into disease models of peripheral organs. A similar approach to the one we used here to isolate rare lung glial cells for scRNA-seq combined with spatial analysis can be extended to characterize the rarer glial sub-populations (e.g. terminal glial cells) and those within other organs as an important step towards identifying the molecular basis for their diversity, functional, and contribution to pathology.

## Materials and methods

### Animals

All animal procedures were performed in accordance with the protocol approved by the Administrative Panel on Laboratory Animal Care (APLAC) at Stanford University. One to seven-month-old mice of the following strains were used: CBA; B6-Tg(Sox10-icre/ERT2) 388Wdr/J (Sox10$^{CreER}$, JAX #027651), C57BL/6N-Gfra3em1(cre/ERT2)Amc/J (Gfra3$^{CreER}$, JAX #029498), B6.Cg-Gt(ROSA)26Sor$^{tm14(CAG-tdTomato)Hze}$/J (Tdtomato, JAX #007914) and Gt (ROSA)26Sor$^{tm1(CAG-EGFP,-mCerulean,-mOrange,-mCherry)Ilw}$ (Rbw, MGI #5441200).

### Ethics statement

All animal husbandry, maintenance, and experiments were performed in accordance with Stanford University's IACUC-approved protocols (APLAC 9780).

### Perfusion and tissue preparation

Mice were euthanized with carbon dioxide and terminally perfused through the right ventricle with 30–60 ml of cold phosphate buffered saline solution (PBS) until the lungs were completely clear of blood. Lungs were collected and drop fixed in 4% paraformaldehyde in PBS for 24 hr, then transferred to 30% sucrose in PBS for 16–24 hr.

**Histology and immunohistochemistry.** Whole mount staining: 300 μm sequential lung sections were prepared using a Compresstome (Precisionary, VF-310-0Z), and compresstome sections were washed with PBS incubated in blocking solution (5% donkey serum/0.5% Triton solution/PBS) overnight at 4˚C. Sections were then incubated for 48 hr at 4˚C in the blocking solution described above with a combination of the following primary antibodies listed in Table 1, and washed with the blocking solution before incubation overnight at 4˚C with corresponding secondary antibodies (Table 2). Sections were incubated with 4′, 6-diamidino-2-phenylindole (DAPI) (1:10,000, ab228549; Abcam, USA) for 1hr, fixed for 1h in 4% paraformaldehyde, washed with PBS, and placed in Cubic 1 solution [42] for imaging. Cryosections: 15–35μm sequential lung sections were prepared using a cryostat (Leica, CM3050S), washed with PBST (PBS/0.03% Triton) and incubated in blocking solution (5% donkey serum/0.03% Triton solution/PBS) for 2h at 4˚C. Sections were then incubated in a combination of the primary antibodies listed in Table 1 for 16–20 hr at 4˚C diluted in blocking solution. After washing, sections were incubated for 45 min at room temperature with corresponding secondary antibodies (see Table 2) diluted in blocking solution. Sections were then incubated in DAPI (1:50,000, ab228549; Abcam, USA) solution diluted in PBS containing 0.03% Triton for the staining of cell nuclei. Sections were mounted with Fluoromount-G (Electron Microscopy Sciences, 17984–25) for imaging.

**Table 1. Primary antibodies for immunohistochemistry.**

| Antigen | Supplier | Catalog No. | Species | Dilution |
|---|---|---|---|---|
| Gfra3 | R&D Systems | AF2645 | Goat Polyclonal | 1:750 |
| GFAP | Dako | Z0334 | Rabbit Polyclonal | 1:1000 |
| Kcna1 | Alomone Labs | APC-009 | Rabbit Polyclonal | 1:1000 |
| S100 | Dako | Z0311 | Rabbit Polyclonal | 1:1000 |
| MBP | Abcam | AB7349 | Rat Monoclonal | 1:200 |
| PGP9.5 | Dako | Z5116 | Rabbit Polyclonal | 1:500 |
| Tuj1 | Covance | MMS-435P | Mouse Monoclonal | 1:750 |
| Tuj1 | BioLegend | 801201 | Mouse Monoclonal | 1:750 |
| TH | EMD Millipore | AB152 | Rabbit Polyclonal | 1:500 |
| CGRP | Abcam | AB36001 | Goat Polyclonal | 1:750 |
| αSMA-FITC conjugated | Sigma | F3777 | Mouse Monoclonal | 1:250 |
| E-cadherin | Invitrogen | 13–1900 | Rat Monoclonal | 1:750 |

*Note.* Gfra3 = GDNF Family Receptor Alpha-3; GFAP = glial fibrillary acidic protein; Kcna1 = Potassium Voltage-Gated Channel Subfamily A Member 1; MBP = myelin basic protein; PGP9.5 = protein gene product 9.5; Tuj1 = beta-III Tubulin; TH = tyrosine hydroxylase; CGRP = Calcitonin Gene Related Peptide; *α*SMA = *α*-smooth muscle actin.

**Multiplex in situ hybridization.** Lung tissues were harvested from *Sox10*[CreER]; *Rosa26*[LSLTdt] mice at postnatal (PN) 28 days old (tamoxifen administered by oral gavage at PN 25 days), fixed in 4% PFA for 24 hours, embedded in OCT, and then cryosectioned at 20–35 μm thickness using a Leica Cm3050s cryostat. Mouse specific RNAscope probes for *Scn7a*, *Cldn19*, *Kcna1*, and *Prx* were designed and produced by Advanced Cell Diagnostics, Inc (ACD) (*Scna7a*, catalog #481921; *Cldn19*, #1292311-C3; *Kcna1*, #481921; *Prx*, #816331-C3). Subsequent steps were performed using the standard protocol RNAscope multiplex fluorescent v2 assay for fixed-frozen samples (UM3231—Rev B) and regents were from the Multiplex Fluorescent Reagent Kit v2 (cat#323270). In brief, sections were first pre-treated during which tissue slides were PBS washed, baked, post-fixed, dehydrated by EtOH, treated by hydrogen peroxide, treated in 99°C RNAscope 1X Target Retrieval Reagent for 5 min, and finally by RNAscope Protease III. Next, sections were hybridized with all three probes for 2h at 40°C, then each of the three RNAscope Multiplex FL v2 AMP for 30 min at 40°C, and finally the HRP step for each of the three RNAscope channels by incubating sections with RNAscope Multiplex FL v2 HRP for 15 min at 40°C, the respective TSA Vivid Dye for 30 min at 40°C,

**Table 2. Secondary antibodies.**

| Secondary Antibody | Conjugated Fluorophore | Supplier | Catalog No. | Dilution |
|---|---|---|---|---|
| Donkey anti-Goat Polyclonal | Alexa Fluor 488 | Invitrogen | A11055 | 1:250 |
| Donkey anti-Goat Polyclonal | Alexa Fluor 647 | Invitrogen | A21447 | 1:250 |
| Donkey anti-Mouse Polyclonal | Alexa Fluor 488 | Invitrogen | A21202 | 1:250 |
| Donkey anti-Mouse Polyclonal | Alexa Fluor 647 | Invitrogen | A31571 | 1:250 |
| Goat anti-Mouse IgG2b | Alexa Fluor 488 | Invitrogen | A21141 | 1:250 |
| Donkey anti-Rat Polyclonal | Alexa Fluor 647 | Jackson Immunos research | 712-605-153 | 1:250 |
| Donkey anti-Rabbit Polyclonal | Alexa Fluor 488 Plus | Invitrogen | A32790 | 1:1000 |
| Donkey anti-Rabbit Polyclonal | Alexa Fluor 647 Plus | Invitrogen | A32795 | 1:1000 |
| Goat anti-Rabbit Polyclonal | Alexa Fluor 488 Plus | Invitrogen | A32731 | 1:1000 |
| Goat anti-Rabbit Polyclonal | Alexa Fluor 647 Plus | Invitrogen | A32733 | 1:1000 |

and the RNAscope Multiplex FL v2 HRP blocker for 15 min at 40˚C. Fluorescence immuno-histochemistry to detect dsRed (to visualize endogenous fluorescent protein expression) was performed as above. Lastly, tissue slides were mounted with Prolong Gold Antifade Mountant containing DPAI and stored in 4˚C before imaging.

**Image acquisition.** Images of immunofluorescent sections were acquired using a Zeiss LSM 880 confocal laser scanning microscope with the Airyscan feature. Tiled images were acquired using a Leica DMI8 Inverted Microscope with 10x, 0.45 numerical aperture objective. Tile scan images were processed using the large volume computational clearing feature in the LAS X software (Leica) to remove background and then merged using the LAS X mosaic merge feature.

**Computational analysis to identify glial cells in whole organism single cell transcriptomic atlases.** For both the Tabula Sapiens and Tabula Muris Senis datasets, data preprocessing was previously performed, which includes scaling the data matrix to 10,000 reads per cell and then performing log normalization. All code generated to analyze the datasets are available at (https://github.com/liangirene/Tabula_Analysis).

*Tabula Muris Senis analysis.* We visualized the expression of classical glial markers including 'Sox10', 'Ncam1', 'Plp1', 'S100b', 'Gfra3', 'Gfap' on a dotplot of the entire data set grouped by the originally annotated leiden clustering (S2A Fig) Each numbered cluster is composed of cells from different tissues. To view the fraction of cells from each originating tissue by cluster, please refer to https://tabula-muris-senis.ds.czbiohub.org/all/facs_droplet/, and highlight the sidebars of 'leiden' and 'tissue'. Based on the expression of these classical markers, we singled out cluster 32 since it had the highest mean expression in the group of the classical and newly identified markers and UMAP projection demonstrates their selective expression (Fig 2A and S2B–S2D Fig).

Differential expression analysis using Wilcoxon rank-sum test for cells in cluster 32 identified three genes, *Gpm6b*, *Scd2*, and *Cryab*, as the top marker genes. p-values from rank-sum testing were Benjamini-Hochberg adjusted. These genes were found to be significantly upregulated in cluster 32 compared to the rest of the cells in the dataset, suggesting that they may play a key role in defining the identity and function of the cells in this cluster. We excluded central nervous system cells (Brain_Non-Myeloid and Brain_Myeloid tissues) from subsequent analysis. Of the 4300 cells in cluster #32 (which represents 1.2% of the total # cells in the entire atlas comprised of 356,213 cells), 3176 cells were from the central nervous system. After filtering these cells, which left us with 1,124 cells, we screened each of the remaining cells for expression of expanded gene panels of classic and newly identified glial cell genes to confirm glial cell identity. In addition, we searched for gene expression of other cell types that could be contaminants (stromal, melanocyte, and epithelial cells) and filtered out those cells co-expressing classic markers of other cell types (S2E and S2F Fig). This left 435 cells of the total 355,213 (0.12%).

Next, we performed Leiden clustering on the remaining cells to identify any potential subpopulation outliers within the cluster (resolution of 0.05). After filtering out the single-cell clusters, we utilized uniform manifold approximation and projection (UMAP) to visualize the remaining clusters. The organs and tissues included for subsequent glial cell analysis were limb muscle, bladder, trachea, fat, heart, kidney, and lung.

**Tabula Sapiens analysis.** Data from *Tabula Sapiens* were imported and a subset of glial cells were identified as detailed here (https://github.com/liangirene/Tabula_Analysis). Glial markers were visualized on a UMAP plot (Fig 5D) to identify clusters most likely to contain glial cells. Differential expression analysis using Wilcoxon rank-sum test with Benjamini-Hochberg correction was performed to identify differentially expressed genes in candidate clusters, and a single candidate cluster was visually selected using the CELLxGENE package.

Subpopulations of cells were manually selected based on both expression of classic glial markers and level of expression of other non-glial markers (Fig 5G). The organs and tissues included after selection using this method were bladder, eye, thymus, tongue, prostate, salivary gland, pancreas, lung and small intestine (S6A Fig). Most of the cells in this cluster were from the large and small intestine of a single donor (donor 14), thus, we excluded these cells from additional analysis at this time because while they expressed common glial genes at high levels, they were also enriched for expression of immediate early genes (*FOS*, *JUN*, *IER2*) suggesting an activated state and were isolated from a single donor.

**Tabula Microcebus Murinus analysis.** For analysis of *Microcebus murinus* data, we extracted myelinating or non-myelinating Schwann cells in peripheral organs (excluding brain and eye). To identify the differentially expressed genes for pan-glia, myelinating, and non-myelinating glia cells in the mouse lemur, we performed Wilcoxon rank-sum test using the 10x data for all genes that were at least moderately expressed in the glia cells (average expression ln $(UMI/10K+1) > 0.1$, $> 5\%$ positive cells). To search for pan-glia cell markers, we further filtered for the genes with higher levels in glia cells compared to the background. The background for pan-glial cell marker calculation is the non-glial cells in the atlas but excluding immune and germ cells (i.e., including epithelial, stromal, endothelial, and non-glia neural). To identify differentially expressed myelinating vs. non-myelinating genes, we compared expression within these two glial cell populations. p-values from the rank-sum test were then Bonferroni adjusted and genes with adjusted p-value $< 0.05$ were shown in S4 Table.

**scRNA-seq data integration.** To integrate the scRNAseq glia data across species, we first aligned genes according to orthology assignments from both the NCBI and Ensembl orthology databases, as described [13] https://www.biorxiv.org/content/10.1101/2021.12.12.469460v2]. This results in a total of ~15,000 one-to-one orthologs across human, mouse lemur, and mouse. Next, we applied Portal [29] to integrate the data and embed the data to a species-integrated UMAP (S6 Table). Parameters for the Portal integration were as follows: integration_sequence = [mouse, lemur, human]; hvg_num = 3000; lambda = [20,20]; training_steps_2000. We then used the species-integrated data to detect species-conserved differentially expressed genes for each of the glia subtype as well as the combination of non-myelinating and terminal glia. We applied right-tailed rank-sum test comparing each of the subtype (or the combination) to the remaining of the glia cells in the dataset. Cells that appear as intermediates of subtypes in the species-integrated UMAP were not counted as the specific subtype but only used as the background in the comparison. Genes with a Bonferroni corrected p-value smaller than 0.05 and a 3-fold increased expression were selected and listed in S7–S10 Tables.

## Supporting information

**S1 Fig. Analysis of lung glial cells by single cell RNA sequencing (scRNA-seq).** (A) Strategy for labeling and isolating lung glial cells using *Ascl1*^CreER^; *Rosa26*^LSL-Zsgreen^ mice using a similar approach we previously reported [17]. Glial cells were enriched by sorting for lineage-labeled EpCAM—/ ZsGreen+ cells by fluorescence-activated cell sorting (FACS). (B) Lower-magnification confocal image of a section (35 μm thick) through an entire mouse lung lobe at E15.5 with glial and neuroendocrine cell progenitors lineage-labeled using (*Ascl1 > ZsGreen1*). Tamoxifen induction between embryonic (E) day 12.5-E13.5 and analysis at E18. E-cadherin (ECAD, white). *Ascl1* –lineage labeled cells (*Ascl1 > Zsgreen1*, green). Scale bar, 200 μm. (C) Close-up view of boxed region in panel B. Four glial cell progenitors (green arrowheads) are located adjacent to airways. Neuroendocrine cells within the airway epithelium (ECAD + regions) are also labeled because they selectively express *Ascl1* within the epithelium. Scale bar, 20 μm. (D) scRNA-seq dot plot showing mean level of expression (dot intensity) and

percent of cells in population with detected expression (dot size) of top glial markers across entire mouse atlas [15]. Cell types: NE, neuroendocrine (PNEC); AT1, alveolar epithelial cell, type 1; AT2, alveolar epithelial cell, type 2; Sm.M, smooth muscle; MyoF, myofibroblast; AdvF, adventitial fibroblast; AlvF, alveolar fibroblast; Peri, pericyte; Meso, mesothelial; Chondro, chondrocyte; Cap-a, capillary aerocyte; Cap, general capillary (Cap-g); Lym, lymphatic cell; B ZBTB32, B cells (ZBTB32+); Reg T, T cells (regulatory); T LY6G5B, T cells (LY6G5B+); NK, natural killer; T Alox5+, T cells (Alox5+); Neut, neutrophil; Baso, basophil; AlvMP, alveolar macrophage; IntMP, interstitial macrophage; pDC, plasmacytoid dendritic; mDC1, myeloid dendritic, type 1; mDC2, myeloid dendritic, type 2; CCR7+ DC, dendritic cell (Ccr7+); Mono (Class), monocyte (classical); Mono(NC), monocyte (non-classical); Mono (Int), monocyte (intermediate).
(TIF)

**S2 Fig. Identification of peripheral glial cells in whole organism single cell transcriptomic atlases of mouse (Tabula Muris Senis).** (A) Dotplot showing expression of classic glial markers (*Sox10*, *Ncam1*, *Plp1*, *S100b*, *Gfra3*, and *Gfap*) across all cell clusters in Tabula Muris Senis. Scale, Log-transformed unique molecular identifiers per 10,000 (ln (UMI)/10K +1). Size of circle indicates fraction of cells in each cluster expressing gene. Scale for all feature plots and heatmaps in this figure represent log-transformed unique molecular identifiers/10,000 +1 [ln (UMI/10K+1)]. (B) UMAP (uniform manifold approximation and projection) plot showing overview of the entire Tabula Muris Senis dataset. Individual clusters are numbered and colored as previously reported for the Tabula Muris Senis dataset (Tabula Muris Consortium, 2020 Nature). Boxed region (red rectangle) indicates cells in a single cluster (cluster #32, n = 4300 cells (1.2%) highly enriched for multiple classic glial genes (C–D) Expression of glial cell selective genes (*Cryab* and *Cdh19*) were identified by marker analysis (see methods section). Central nervous system (CNS) glial cells (n = 3176 cells) were removed from subsequent analysis. Scale, expression level. (E-G) Heatmaps showing expression of pan glial genes (*Sox10*, *Plp1*, *S100b*, *Ncam1*), the classic glial marker gene (*Gfap*), myelinating genes (*Pllp*, *Mpz*, *Mbp*, *Pmp22*, *Cdkh1c*) and non-myelinating genes (*Scn7a*, *Gfra3*, *Apoe*, *Nrxn1*) across each of the 7 organs in *Tabula Muris* in which we identified glial cells (indicated by light orange color bars above heatmap). Non-glial cells (grey color bar) were removed from the data set for a final group of 435 glial cells. Note that most of the kidney cells (G) expressed epithelial genes, *Epcam* and *Cdh1*, but not pan-glial genes. Thus, they were removed from further analysis. (TIF)

**S3 Fig. Anatomic distribution of *Sox10* + cells in mouse lung.** (A) Fluorescence microscope image of a proximal airway (computationally cleared, THUNDER Imager, Leica). *Sox10*-lineage-labeled glial cells (*Sox10$^{CreER}$ > Tdt*) form a mesh-like network along airways, but are absent from pulmonary arteries. Scale bar, 100 μm. αSMA, alpha smooth muscle actin. Pulmonary artery (pulm. artery) (B) Representative confocal image of a second airway showing NMG cells forming a networks along the airway surface closely associated with nerve fibers (protein gene produce 9.5/PGP9.5, white). Airway smooth muscle fibers (alpha smooth muscle actin/α-SMA, green). Scale bar, 100 μm (C) Representative sagittal section of distal airway surrounded by a network of *Sox10*-expressing glia (*Sox10$^{CreER}$>Tdt*) and adjacent alveolar regions. Ecad, E-cadherin. Scale bar, 50 μm. (D) Confocal image of a representative section of pulmonary vein (pulm. vein) with *Sox10$^{CreER}$; Tdt* labeled glial cells co-localizing with TH+ neuronal fibers. Alpha-smooth muscle actin, α−SMA. Bar, 50 μm. (E and E') Individual channels highlighting boxed region in panel B showing *Sox10$^{CreER}$>Tdt* lineage labeled NMG cells as individual channels shown for boxed region. TH, tyrosine hydroxylase. Bar, 50 μm. (F) Quantification of *Sox10*+ glial cells associated with TH+ nerves in myelinating (n = 247/267,

92.5% cells), non-myelinating (n = 1081/1623, 66.6%) airway glial cells vs. glial cells (n = 83/83, 100%) of pulmonary veins. Statistically significant differences were observed between airway MG vs. NMG cells (p < 0.0001) and between non-myelinating glial (NMG) cells of airway (A) vs. pulmonary vein (PV) (p < 0.0001). Two-proportion z-test. Error bars, 95% C.I. indicated.
(TIF)

**S4 Fig. Diverse cellular morphologies of lung glia revealed by in vivo single cell labeling.** (A-B) Labeling of single non-myelinating glial cells along the airway using $Sox10^{CreER}$ mice in combination with the $Rosa26^{LSL-Rbw}$ multicolored reporter ($Sox10^{CreER} > Rbw$). Two individual NMG cells along the airway representing (A) unipolar and (B) bipolar cells and their projections. Scale 10 μm. (C-D) Single multipolar glial cells labeled using $Gfra3^{CreER}$; $Rosa26^{LSL-Tdt}$ mice. Scale 10 μm. Sparse cell labeling was observed in glial cells with $Gfra3^{CreER}$ driving recombination or cells expressing the $Rosa26^{LSL-Rbw}$ reporter, which permitted visualization of single cell morphologies. Schematics showing fine extensions corresponding to each panel (below) (E) Quantification of $Sox10$ lineage+ glial cells by morphology of unipolar (n = 3/84 cells in pulmonary vein; 1/115 cells in airway), bipolar (45/84 cells in vein; 60/115 cells in airway), and multipolar (36/84 cells in vein; 54/115 cells in airway). Age of mice: PN 78 and 101 days, n = 2 mice. Grey bars, pulmonary vein (Pulm.vein); black bars, airway. Statistically significant differences were observed between unipolar vs bipolar, airway (p<0.0001), unipolar vs multipolar, airway (p<0.0001), unipolar vs bipolar, vein (p<0.0001), and unipolar vs multipolar, vein (p<0.0001). 2-proportion z test. Error bars, 95% C.I. (F) Basal $TG^{NS}$ cell in $Sox10^{CreER} > Tdt$ mice with $TG^{NS}$ soma (indicated by white arrowhead and outlined by white dotted lines) associated with a neuroepithelial body (NEB) and lacks GFRA3 expression (green) by immunohistochemistry. PGP9.5 (NEBs and neurons, white). Scale 10 μm.
(TIF)

**S5 Fig. Differentially expressed genes in myelinating vs. non-myelinating glial cells in the whole organism mouse lemur scRNA-seq atlas (Tabula Microcebus Murinus).** (A) Heatmap showing scRNA-seq expression levels of differentially expressed genes in the myelinating and non-myelinating glia identified in bladder, bone, lung, tongue, fat, kidney, pancreas, and small intestine from Tabula Microcebus Murinus [13]. The glial cells in Tabula Microcebus Murinus were annotated by expression of classic glial genes, recently reported genes in peripheral glia, and those we identified in our analysis of Tabula Muris Senis data. Relative expression levels are shown, normalized to the robust maximal (max.) (99th percentile) expression of the gene among all glia cell types.
(TIF)

**S6 Fig. Identification of peripheral glial cells in whole organism single cell transcriptomic atlases of human (Tabula Sapiens).** (A) UMAP plots representing glial cells isolated from each organ prior to filtering out the intestinal glial cells as previously described (also see methods) (B) Expression of pan-glial genes ($SOX10$, $PLP1$, $NCAM1$, $GPM6B$, $CDH19$, and $CRYAB$), and conserved expression of non-myelinating gene ($SCN7A$).
(TIF)

**S7 Fig. Conserved expression of apolipoprotein E ($Apoe$) in non-myelinating glial cells across organs in mouse and mouse lemur.** (A) UMAP representation of glial cells identified in mouse atlas with myelinating and non-myelinating glial cells from 7 different organs (Fig 3B and 3C). Scale. Values are log transformed unique molecular identifiers (UMI) per 10,000, ln (UMI/10K +1) (B) Dot plot showing expression of $Apoe$ expression by scRNA-seq across all cells of the mouse lung cell atlas [15] across all major compartments (epithelial, stromal,

vascular, and immune). Low level *Apoe* expression was observed in AT2 cells, and within the conducting airway epithelium, only basal cells expressed *Apoe* under basal conditions. Prominent expression within interstitial macrophages populations was observed. In experimental mouse models of allergic inflammation and ex vivo assays of human macrophages isolated from bronchoalveolar fluid, APOE secretion could be induced in a predominantly alveolar macrophage population [33]. Our current analysis reveals minimal expression in alveolar macrophages under basal conditions, suggesting an alternate molecularly defined population of macrophages that expresses *APOE* under basal conditions. Expression of glial markers (*Gfap*, *Gfra3*, *Sox10*). Expression levels of epithelial genes keratin 5/*Krt5*, EPCAM/*Epcam* and Cadherin-1/*Cdh1* are shown. (C) Heatmap showing conserved expression of *APOE* in mouse lemur glial cells across the 8 organs represented (tongue; small intestine; S.intestine; pancreas; lung; kidney; fat; bone; and bladder). Relative expression levels are shown, normalized to the robust maximal (99$^{th}$ percentile) expression of the gene among all glia cell types.
(TIF)

**S8 Fig. Cellular and molecular features of *Sox10*+ lineage-labeled glial cells of bladder.** (A) Single plane cross-section through bladder showing prominent glial network in the outer detrusor muscle (DM) layer and a sparser network in the subepithelial lamina propria (LP) layer. E-cadherin (ECAD) expression is detected by fluorescence immunohistochemistry. Bar, 100 μm. (B) Close-up confocal single plane images of two non-myelinating Sox10 lineage positive (*Sox10^{CreER}* > Tdt) glial cells in the detrusor muscle. Bar, 10 μm. (C) Close up images of myelinating glial cells running parallel to nerve fibers at the periphery of the bladder. Myelinated nerve fibers expressing myelin basic protein (MBP) indicated in adjacent panel. Bar, 10 μm. Note the absence of GFRA3 expression in the myelinating cells consistent with the scRNA-seq data. (D) Quantification of lineage-labeled (*Sox10^{CreER}* > Tdt) glial cells co-expressing myelinating glial marker, myelin basic protein (MBP) in the detrusor muscle (DM) vs. the lamina propria (LP). MBP production by IHC: lamina propria (0/136, 0%); detrusor muscle (41/375, 11%). A statistically significant difference between the lamina propria and the detrusor muscle was observed (p<0.0001, 2 proportion z test). Error bars, 95% C.I. indicated. (E) Quantification of lineage-labeled (*Sox10^{CreER}* > Tdt) glial cells co-expressing classic glial markers (S100B, GFAP) and GFRA3 in the detrusor muscle (DM) vs. the lamina propria (LP). S100 expression by immunohistochemistry (IHC): lamina propria (43/62, 69%); detrusor muscle (113/190, 59%). GFAP: lamina propria (1/46, 2%); detrusor muscle (13/210, 6%). GFRA3: lamina propria (78/83, 94%); detrusor muscle (69/186, 37%). A statistically significant difference between the lamina propria and the detrusor muscle was observed in GFRA3 (p<0.0001, 2 proportion z test). Error bars, 95% C.I. indicated.
(TIF)

**S1 Table. Lung glial cell markers in mouse.**
(XLSX)

**S2 Table. *Tabula Muris Senis* glial cell metadata.**
(XLSX)

**S3 Table. Differentially expressed glial genes in mouse peripheral glial (Tabula Muris Senis).**
(XLSX)

**S4 Table. Differentially expressed glial genes (*Tabula Microcebus*).**
(XLSX)

**S5 Table. *Tabula Sapiens* glial cell scRNA-seq metadata.**
(XLSX)

**S6 Table. Metadata for peripheral glial cells in integrated dataset.**
(XLSX)

**S7 Table. Differentially expressed myelinating glial (MG) cell genes in integrated dataset.**
(XLSX)

**S8 Table. Differentially expressed non-myelinating glial (NMG) cell genes in integrated dataset.**
(XLSX)

**S9 Table. Differentially expressed terminal glial (TG) cell genes in integrated dataset.**
(XLSX)

**S10 Table. Differentially expressed non-myelinating glial (NMG) cell genes in integrated dataset (including TG).**
(XLSX)

**S11 Table. Summary of peripheral glial cell classes identified across organs and species.**
(DOCX)

**S1 Movie. Satellite glial cells associated with intrinsic ganglia of lung.**
(MP4)

**S2 Movie. Terminal glial cell with projection terminating within a neuroepithelial body (NEB).**
(MP4)

## Acknowledgments

We thank Parth Arora for assistance with the in situ hybridization experiment and Nicole Almanzar for assisting with the morphologic characterization of mouse lung glial cells and providing comments on the manuscript. We thank Professor Mark Krasnow and members of the Krasnow Lab for discussions, and John Whitin (Director of Laboratory Research Development, Department of Pediatrics) for assisting with our research lab move into a new building in the month before pandemic shutdowns. C.S.K. is a Tashia and John Morgridge Endowed Faculty Scholar of the Maternal and Child Health Research Institute (MCHRI).

## Author Contributions

**Conceptualization:** Christin S. Kuo.

**Data curation:** Christin S. Kuo.

**Formal analysis:** Shaina Hall, Shixuan Liu, Irene Liang, Shawn Schulz, Camille Ezran, Christin S. Kuo.

**Funding acquisition:** Christin S. Kuo.

**Investigation:** Shaina Hall, Mingqian Tan, Christin S. Kuo.

**Methodology:** Christin S. Kuo.

**Project administration:** Christin S. Kuo.

**Resources:** Christin S. Kuo.

**Software:** Shixuan Liu, Irene Liang, Shawn Schulz.

**Supervision:** Christin S. Kuo.

**Validation:** Christin S. Kuo.

**Visualization:** Shixuan Liu, Christin S. Kuo.

**Writing – original draft:** Shaina Hall, Shixuan Liu, Camille Ezran, Christin S. Kuo.

**Writing – review & editing:** Shaina Hall, Shixuan Liu, Camille Ezran, Christin S. Kuo.

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
