## [Decision Letter · Decision Letter 0]

6 Aug 2024

PONE-D-24-20617Cellular and molecular characterization of peripheral glia in the lung and other organsPLOS ONE

Dear Dr. Kuo,

Thank you for submitting your manuscript to PLOS ONE. After careful consideration, we feel that it has merit but does not fully meet PLOS ONE’s publication criteria as it currently stands. Therefore, we invite you to submit a revised version of the manuscript that addresses the points raised during the review process.

We look forward to receiving your revised manuscript.

Kind regards,

Kenji Tanigaki, Ph.D., M.D.

Academic Editor

PLOS ONE

Journal Requirements:

"Chan Zuckerberg Initiative, Human Lung Cell Atlas (60053823 SU) and Stanford University COVID relief program. C.S.K. is a Tashia and John Morgridge Endowed Faculty Scholar of the Maternal and Child Health Research Institute (MCHRI). "

"We thank Nicole Almanzar for contributing to the morphologic characterization of mouse lung glial cells and for providing comments on the manuscript and Parth Arora for assistance with the in situ hybridization experiment. We thank members of the Krasnow Lab for discussions. This work was supported by funding from Chan Zuckerberg Initiative, Human Lung Cell Atlas (60053823 SU) and Stanford University COVID relief program. We thank Mark Krasnow for funding acquisition and support for this project. We also thank Stanford University School of Medicine, Department of Pediatrics for resources. C.S.K. is a Tashia and John Morgridge Endowed Faculty Scholar of the Maternal and Child Health Research Institute (MCHRI). "

"Chan Zuckerberg Initiative, Human Lung Cell Atlas (60053823 SU) and Stanford University COVID relief program. C.S.K. is a Tashia and John Morgridge Endowed Faculty Scholar of the Maternal and Child Health Research Institute (MCHRI). "

6. Please include a copy of Tables 1 and 2 which you refer to in your text on page 16.

Reviewers' comments:

Reviewer's Responses to Questions

**Comments to the Author**

1. Is the manuscript technically sound, and do the data support the conclusions?

Reviewer #1: Yes

Reviewer #2: Yes

2. Has the statistical analysis been performed appropriately and rigorously? 

Reviewer #1: Yes

Reviewer #2: Yes

3. Have the authors made all data underlying the findings in their manuscript fully available?

Reviewer #1: Yes

Reviewer #2: Yes

4. Is the manuscript presented in an intelligible fashion and written in standard English?

Reviewer #1: Yes

Reviewer #2: Yes

5. Review Comments to the Author

Reviewer #1: Hall et al. systematically studied the transcriptomic and anatomic diversity of peripheral glia, an important but under-studied cell population. These cells are challenging to study as they are rare cell types and are hard to isolate. Combining mouse genetics, scRNA-seq (both in-house generated data and public data), and several imaging and data analysis tools, they showed that peripheral glia contains 3 subtypes – MG, NMG and TG with distinct and conserved molecular signatures across species and distinct morphological features. Overall, the study is well-designed and conducted, the data is convincing, and the manuscript is of interest to readers who are interested in peripheral glia and who are interested in identifying rare cell types using scRNA-seq data.

I have no major comments. below are some minor comments:

In Figure 5C Tabula Microcebus data (mouse lemur), pan-glial markers seem to be weak, compared with the mouse and human data (the same heatmaps in Figure 2I and Figure 5H. Could the authors briefly explain the reasons?

In Figure 1E: please label all cell types in the violin plots. It is up to the authors to see if it is also useful to include marker genes for other cell types.

In Figure 2/5/6 and many supp figures: please clarify gene expression scale. “ln(UMI/K+1)” shows up in many main figures, “ln(UMI/10K+1)” shows up in many supplementary figures, and “ln(CP10K+1)” shows up sometimes. According to the text description, “ln(UMI/K+1)” may not be correct. The other two should mean the same thing, and either one is fine.

Some supplementary tables are off — they need to be correctly ordered and referenced. Below are a few (maybe incomplete) instances:

In Methods “p-values from the rank-sum test were then Bonferroni adjusted and genes with adjusted p-value < 0.05 were shown in Table X”

“Next, we applied Portal [29] to integrate the data and embed the data to a species-integrated UMAP (Table S7)”, which should be table S6?

“Genes with a Bonferroni corrected p-value smaller than 0.05 and a 3-fold increased expression were selected and listed in Table S6.”, which does not contain DEGs information.

In the Supplementary table list “S2 Table. Differentially expressed glial genes in mouse peripheral glial (Tabula Muris Senis)”, which should be Table S3?

Also in tables describing DEGs results, sometimes there are both effect sizes (fold change) and adjusted p-values. Sometimes it has only p-values, without effect size and without indicating whether or not p-values are adjusted. This should be made consistent.

Reviewer #2: Peripheral glial cells play crucial roles in regulating distinct physiological functions in peripheral organs. However, the molecular signatures and spatial locations of peripheral glial cell types are not fully understood. Hall and colleagues used scRNA-seq to characterize the cellular and molecular features of peripheral glial cells. They identified three distinct populations and utilized immunostaining and in situ hybridization to characterize them. They also compared their data with published atlases and found that these glial cell types are conserved across species. This study provides valuable and important information about peripheral glial cells, contributing to the field of peripheral glia.

I have a few comments that might help the authors revise this manuscript:

Figure 1: It would be great if the authors could provide a more comprehensive confocal image showing the labeling of Ascl1

; Rosa:zsgreen in the lung, perhaps a low-magnification image.

Figures 1D-E: Please also show the expression patterns of canonical markers for other cell types in the feature plots or violin plots.

Figure 2A: This figure is not cited in the text, and it is unclear why the authors present the expression pattern of Plp1. Is it a pan-glial marker? It would make more sense to show the expression patterns of pan-glial markers in the atlas they analyzed in the main figure.

Figure S2A-D: These figures are not cited. In Figure S2, please add a UMAP plot with the corresponding cluster numbers labeled. Also, the text states that 435 glial cells were found, composing 0.12% of the atlas. Why does it show 4300 cells, 1.2%, in Figure S2B?

Differential Expression (DE) Analysis: It would be interesting to perform DE analysis among the three glial types and investigate their molecular signatures.

Gfra3 Expression: It is interesting to note that Gfra3 is selectively expressed in NMG. How about other growth factor receptors?

Figure 6: Again, it would be interesting to see the molecular signatures of these three glial cell types.

There are some grammar issues and typos. For instance,

“we used the expression profiles from both the Tabula Muris Senis and Ascl1-enriched lung scRNA-seq datasets described above to predicted that” -> “to predict that”

“Individual non-myelinating airway and vascular glial cells were often had multipolar projections compared to myelinating glia cells” -> “glial cells often had”

6. PLOS authors have the option to publish the peer review history of their article (what does this mean?). If published, this will include your full peer review and any attached files.

Reviewer #1: **Yes: **Fangming Xie

Reviewer #2: No

---

## [Author Response · Author response to Decision Letter 0]

21 Aug 2024

Dear Dr. Tanigaki,

Thank you for handling our manuscript and for the constructive comments and suggestions and reviews. Please see our revisions and responses (bold type) to each point below. This is summarized in the attached 'rebuttal' letter entitled, 'Response to Reviewers'.

---

## [Editor Report · Decision Letter 1]

29 Aug 2024

Cellular and molecular characterization of peripheral glia in the lung and other organs

PONE-D-24-20617R1

Dear Dr. Kuo,

We’re pleased to inform you that your manuscript has been judged scientifically suitable for publication and will be formally accepted for publication once it meets all outstanding technical requirements.

Kind regards,

Kenji Tanigaki, Ph.D., M.D.

Academic Editor

PLOS ONE
---

## [Editor Report · Acceptance letter]

4 Sep 2024

PONE-D-24-20617R1 

PLOS ONE

Dear Dr. Kuo, 

I'm pleased to inform you that your manuscript has been deemed suitable for publication in PLOS ONE. Congratulations! Your manuscript is now being handed over to our production team.

Kind regards, 

on behalf of

Dr. Kenji Tanigaki 

Academic Editor

PLOS ONE